# Comparative transcriptomics reveal a novel tardigrade-specific DNA-binding protein induced in response to ionizing radiation

Marwan Anoud[1,2†], Emmanuelle Delagoutte[1†], Quentin Helleu[1†], Alice Brion[1], Evelyne Duvernois-Berthet[3], Marie As[1], Xavier Marques[1,4], Khadija Lamribet[1], Catherine Senamaud-Beaufort[5], Laurent Jourdren[5], Annie Adrait[6], Sophie Heinrich[7,8], Geraldine Toutirais[9], Sahima Hamlaoui[10], Giacomo Gropplero[11], Ilaria Giovannini[12,13], Loic Ponger[1], Marc Geze[4], Corinne Blugeon[5], Yohann Couté[6], Roberto Guidetti[12,13], Lorena Rebecchi[12,13], Carine Giovannangeli[1], Anne De Cian[1]*, Jean-Paul Concordet[1]*

[1]Département AVIV, MNHN, CNRS UMR7196, INSERM U1154, Paris, France; [2]Université Paris-Saclay, Orsay, France; [3]Département AVIV, MNHN, CNRS UMR7221, Paris, France; [4]CeMIM, MNHN, CNRS UMR7245, Paris, France; [5]Génomique ENS, Institut de Biologie de l'ENS (IBENS), Ecole Normale Supérieure, CNRS, INSERM, Université PSL, Paris, France; [6]Univ. Grenoble Alpes, INSERM, CEA, UA13 BGE, CNRS, CEA, Grenoble, France; [7]Institut Curie, Inserm U1021-CNRS UMR 3347, Université Paris-Saclay, Université PSL, Orsay Cedex, France; [8]Plateforme RADEXP, Institut Curie, Orsay, France; [9]Plateforme technique de Microscopie Electronique, MNHN, Paris, France; [10]Département AVIV MNHN, UMR7245, Paris, France; [11]MMBM, Institut Curie, CNRS UMR168, Paris, France; [12]Department of Life Sciences, University of Modena and Reggio Emilia, Modena, Italy; [13]NBFC, National Biodiversity Future Center, Palermo, Italy

*For correspondence:
anne.de-cian@mnhn.fr (ADC);
jean-paul.concordet@mnhn.fr
(J-PC)

†These authors contributed
equally to this work

**Competing interest:** The authors
declare that no competing
interests exist.

Reviewing Editor: Yamini Dalal,
National Cancer Institute, United
States

**Abstract** Tardigrades are microscopic animals renowned for their ability to withstand extreme conditions, including high doses of ionizing radiation (IR). To better understand their radio-resistance, we first characterized induction and repair of DNA double- and single-strand breaks after exposure to IR in the model species *Hypsibius exemplaris*. Importantly, we found that the rate of single-strand breaks induced was roughly equivalent to that in human cells, suggesting that DNA repair plays a predominant role in tardigrades' radio-resistance. To identify novel tardigrade-specific genes involved, we next conducted a comparative transcriptomics analysis across three different species. In all three species, many DNA repair genes were among the most strongly overexpressed genes alongside a novel tardigrade-specific gene, which we named *Tardigrade DNA damage Response 1* (*TDR1*). We found that TDR1 protein interacts with DNA and forms aggregates at high concentration suggesting it may condensate DNA and preserve chromosome organization until DNA repair is accomplished. Remarkably, when expressed in human cells, TDR1 improved resistance to Bleomycin, a radiomimetic drug. Based on these findings, we propose that TDR1 is a novel tardigrade-specific gene conferring resistance to IR. Our study sheds light on mechanisms of DNA repair helping cope with high levels of DNA damage inflicted by IR.

## eLife assessment

This study offers **valuable** insight into the remarkable resistance of tardigrades to ionizing radiation by showing that radiation treatment induces a suite of DNA repair proteins and by identifying a strongly induced tardigrade-specific DNA-binding protein that can reduce the number of double-strand breaks in human U2OS cells. The evidence of upregulation of repair proteins is **convincing**, and the case for a role of the newly identified protein in repair can be strengthened as genetic tools for tardigrades become better developed. The results will interest the fields of DNA repair and radiobiology as well as tardigrade biologists.

## Introduction

Tardigrades are microscopic animals found in marine or freshwater environments, and in semi-terrestrial habitats such as moss, lichen, and leaf litter. They are well known for their resistance to IR (*Jönsson, 2019*) and extreme conditions like desiccation, freezing, and osmotic stress (*Guidetti et al., 2011*). With over 1400 species, they belong to the clade of ecdysozoans, which also includes nematodes and arthropods (*Degma and Roberto, 2023*). Tardigrades share a highly conserved body plan, with a soft body protected by a cuticle, four pairs legs, and a characteristic feeding apparatus. Tardigrades, however, can differ in their resistance to extreme conditions. For example, *Ramazzottius oberhaeuseri* withstands extremely rapid desiccation while *Hypsibius dujardini* only survives gradual dehydration (*Wright, 1989*), and the freshwater *Thulinius ruffoi* is not resistant to desiccation (*Kondo et al., 2020*). Many species across the Tardigrada phylum can tolerate irradiation doses higher than 4000 Gy (*Hashimoto and Kunieda, 2017*), but some are less tolerant, like *Echiniscoides sigismundi*, which has an LD50 at 48 hr of 1000 Gy (*Jönsson et al., 2016*). However, the doses compatible with maintenance of fertility seem much lower, e.g., the maximum is 100 Gy for *Hypsibius exemplaris* (*Beltrán-Pardo et al., 2015*). Due to challenges in rearing tardigrades in the laboratory (*Altiero and Rebecchi, 2001*), the maintenance of fertility has seldom been investigated and some species might remain fertile at higher doses.

Understanding the genes involved in tardigrade resistance to IR is essential to unraveling the mechanisms of their exceptional resilience. Systematic comparison of whole-genome sequences has suggested that tardigrades have one of the highest proportions of gene gain and gene loss among metazoan phyla (*Guijarro-Clarke et al., 2020*). Several novel, tardigrade-specific genes have indeed been involved in resistance to desiccation including CAHS, MAHS, SAHS, and AMNP gene families (*Hesgrove and Boothby, 2020*; *Arakawa, 2022*; *Yamaguchi et al., 2012*; *Tanaka et al., 2015*; *Yoshida et al., 2022*). For resistance to IR, the tardigrade-specific gene Dsup (for DNA damage suppressor) has been discovered in *Ramazzottius varieornatus*. Dsup encodes an abundant chromatin protein that increases resistance to X-rays when expressed in human cells (*Hashimoto et al., 2016*). In vitro experiments have shown that DNA damage induced by hydroxyl radicals was reduced when Dsup was added to nucleosomal DNA (*Chavez et al., 2019*), indicating DNA protection by Dsup. However, it is not yet possible to inactivate genes with CRISPR-Cas9 in tardigrades (*Goldstein, 2022*) and direct evidence for the importance of Dsup in radio-resistance of tardigrades is still lacking. Interestingly, the presence of resistance genes differs across tardigrade genomes (*Arakawa, 2022*). While AMNP genes are found in both classes of tardigrades, Heterotardigrada and Eutardigrada, CAHS, SAHS, and MAHS genes are only found in Eutardigrada, and Dsup appears restricted to the Hypsibioidea superfamily of Eutardigrada (*Arakawa, 2022*). However, given the range of species demonstrated to be radio-resistant across the phylum (*Hashimoto and Kunieda, 2017*), it seems likely that additional tardigrade-specific genes are involved in tardigrades' radio-resistance.

In addition to tardigrades, other animals display exceptional resistance to IR including rotifers, nematodes, and larvae of *Polypedilum vanderplanki* midges, all surviving doses more than 100 times higher than humans. Recent studies have begun to shed light on the mechanisms involved (*Ujaoney et al., 2024*). In addition to DNA protection, DNA repair may also help maintain genome integrity upon irradiation. In rotifers, the rate of DNA double-strand breaks (DSBs) is equivalent to that in human cells (*Gladyshev and Meselson, 2008*), showing that DNA repair, rather than DNA protection, plays a predominant role in their radio-resistance. Furthermore, it was recently found that genes of DNA repair are upregulated in response to IR in rotifers and *P. vanderplanki* larvae (*Moris et al., 2023*; *Ryabova et al., 2017*). In rotifers, upregulation of a DNA ligase gene acquired by horizontal gene transfer may be essential to radio-resistance (*Nicolas et al., 2023*). In prokaryotes, radio-resistance

has been investigated in the bacterium *Deinococcus radiodurans*, showing highly efficient DNA repair in response to the high levels of DNA damage induced by high doses of IR and the contribution of *D. radiodurans* DNA repair genes (*Timmins and Moe, 2016*). Previous studies have suggested that upregulation of DNA repair genes may also play a role in radio-resistance of tardigrades: irradiation with IR increases expression of Rad51, the canonical recombinase of homologous recombination (HR), in *Milnesium inceptum* (*Beltrán-Pardo et al., 2013*), and regulation of DNA repair genes was observed in *R. varieornatus* (*Yoshida et al., 2021*).

To improve our understanding of resistance to IR in tardigrades, we sought to characterize DNA damage and repair after irradiation and to identify novel tardigrade-specific genes involved in resistance to IR. For this purpose, we first examined the kinetics of DNA damage and repair after IR in the model species *H. exemplaris*. This species was chosen due to its ease of rearing in laboratory conditions and its known genome sequence. Additionally, to identify novel genes involved in resistance to IR, we analyzed gene expression in response to IR in *H. exemplaris* and two additional species, *Acutuncus antarcticus,* from the Hypsibioidea superfamily (*Giovannini et al., 2018*), and *Paramacrobiotus fairbanksi* of the Macrobiotoidea superfamily (*Guidetti et al., 2019*). Together with multiple DNA repair genes, a tardigrade-specific gene, which we named *T*ardigrade *D*NA damage *R*esponse gene 1 (TDR1), was strongly upregulated in response to IR in all three species analyzed. Further analyses in *H. exemplaris*, including differential proteomics and western blots, showed that TDR1 protein is present and upregulated. In vitro experiments demonstrated that recombinant TDR1 interacts with DNA and forms aggregates with DNA at high concentrations. Importantly, when expressed in human cells, TDR1 reduced the number of phospho-H2AX foci induced by Bleomycin, a DNA damaging drug used as a radiomimetic. These findings show the importance of DNA repair in radio-resistance of tardigrades and suggest that TDR1 is a novel tardigrade-specific DNA-binding protein involved in DNA repair after exposure of tardigrades to IR.

## Results

### DSBs and SSBs are induced and repaired after exposure of *H. exemplaris* to IR

IR causes a variety of damages to DNA such as nucleobase lesions, single-strand breaks (SSBs) and DSBs (*Téoule, 1987*). In eukaryotes, from yeast to humans, phosphorylation of H2AX is a universal response to DSBs and an early step in the DNA repair process (*Fernandez-Capetillo et al., 2004*). To investigate DSBs caused by IR, we generated an antibody against phosphorylated H2AX of *H. exemplaris* (*Figure 1—figure supplement 1*). *H. exemplaris* tardigrades were exposed to either 100 Gy or 1000 Gy of $^{137}$Cs γ-rays, which are known to be well tolerated by this species (*Beltrán-Pardo et al., 2015*). We analyzed phospho-H2AX in protein extracts of *H. exemplaris* collected at 30 min, 4 hr, 8h30, 24 hr, and 73 hr after irradiation. For both 100 Gy and 1000 Gy doses, phospho-H2AX was detected at 30 min after irradiation, reached its peak levels at 4 hr and 8h30 and then gradually decreased (*Figure 1a*). Irradiation was also performed with an accelerated electron beam, which delivered identical doses in much shorter times, 1000 Gy in 10 min instead of 1 hr for the $^{137}$Cs source, in order to better appreciate the early peak of phospho-H2AX. A peak of phospho-H2AX was detected at 4 hr and a similar, gradual decrease was observed (*Figure 1—figure supplement 2a*). Next, we performed whole-mount immunolabeling of tardigrades and observed intense, ubiquitous phospho-H2AX labeling in nuclei 4 hr after 100 Gy irradiation, which had significantly decreased 24 hr later (*Figure 1b*). This suggests irradiation impacts all adult cells and indicates efficient DNA repair by 24 hr after 100 Gy irradiation, consistent with the results of western blot analysis. After 1000 Gy irradiation, the intense signal detected at 4 hr had decreased in most nuclei at 24 hr but it persisted at high intensity specifically in gonads (*Figure 1—figure supplement 2b and c*). The finding of persistent DSBs in gonads at 72 hr after 1000 Gy likely explains why *H. exemplaris* no longer lay eggs and become sterile after exposure to 1000 Gy (*Beltrán-Pardo et al., 2015*). In order to investigate DNA synthesis taking place after irradiation, we incubated tardigrades with the thymidine nucleotide analog EdU (*Gross et al., 2018*). Using confocal microscopy, we could detect DNA synthesis in replicating intestinal cells of control animals, as previously shown by *Gross et al., 2018*. In contrast, we could not detect any specific signal in irradiated tardigrades compared to controls, suggesting (i) that DNA synthesis induced during DNA repair remained at low, undetectable levels and (ii) that dividing intestinal cells

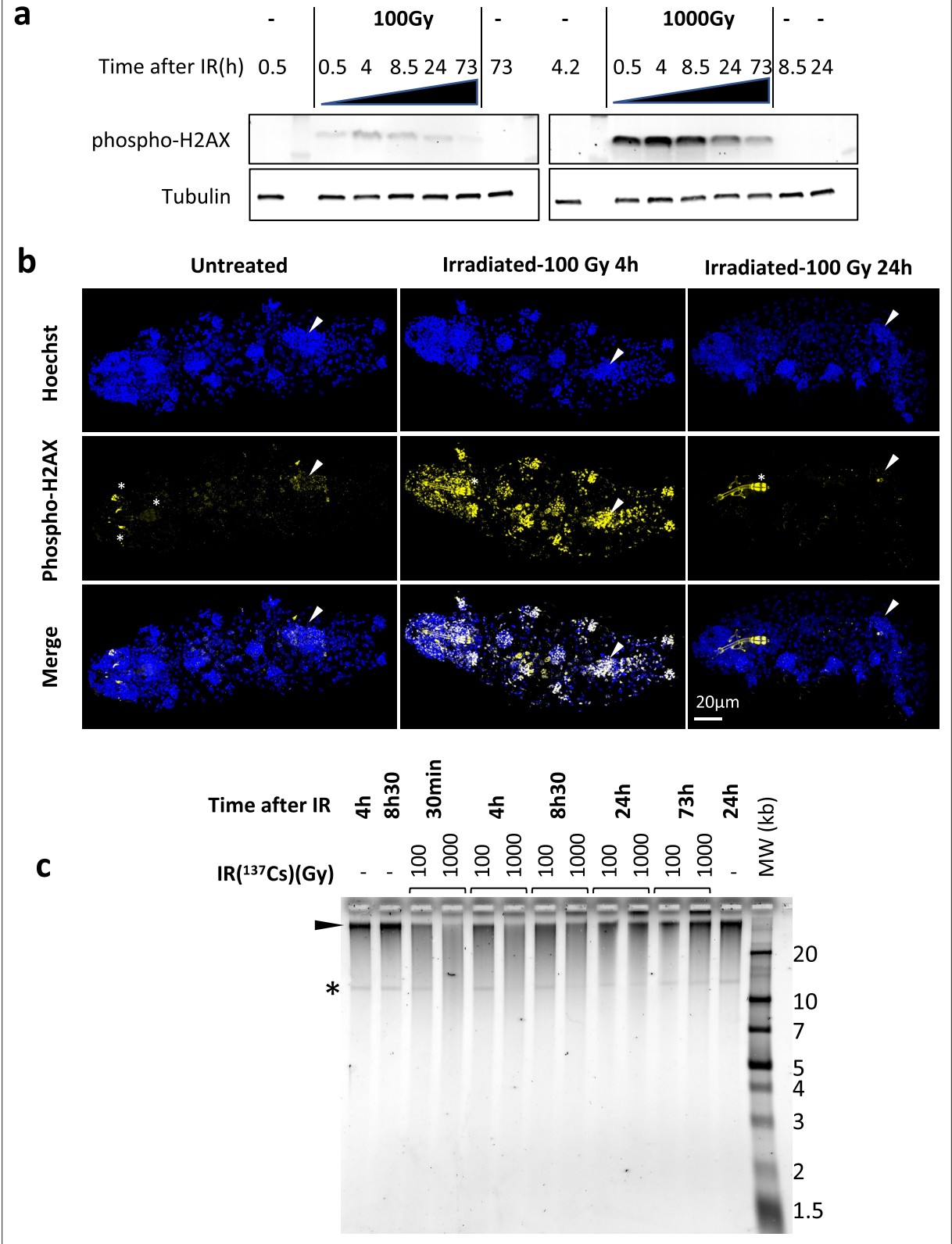

**Figure 1.** Analysis of DNA damage and repair in *H. exemplaris* after γ-ray irradiation. (**a**) Analysis of phospho-H2AX expression after exposure of *H. exemplaris* to ionizing radiation (IR). Western blot analysis with in-house antibody against phosphorylated *H. exemplaris* H2AX (anti-phospho-H2AX) at indicated time points after irradiation of tardigrades with indicated dose of γ-ray irradiation. Phospho-H2AX levels were normalized by total alpha-tubulin expression levels and quantification is provided in *Figure 1—figure supplement 2a*. (-) lanes show extracts from control tardigrades processed

*Figure 1 continued on next page*

*Figure 1 continued*

in parallel to irradiated tardigrades at indicated time points post-irradiation. (**b**) Analysis of phospho-H2AX expression in whole-mount *H. exemplaris* after exposure to 100Gy. Tardigrades were exposed to 100Gy, fixed with 4% PFA at 4 hr and 24hr post-irradiation, immunolabeled with anti-phospho-H2AX antibody and anti-rabbit IgG conjugated to Alexa Fluor 488 and visualized by confocal microscopy using the Airyscan2 module. Maximum projection of confocal Z-stack are shown. Images at different time points were taken with identical settings so that signal intensity could be compared. Upper panel shows Hoechst staining of nuclei (in blue). Arrowhead indicates position of the gonad (revealed by intense Hoechst and larger nuclei signal). The gonad exhibits intense labeling phospho-H2AX at 4hr which is no longer detected at 24hr, showing efficient DNA repair consistent with preservation of the capacity to lay eggs and reproduce after 100Gy IR (*Beltrán-Pardo et al., 2015*). * indicates autofluorescence of bucco-pharyngeal apparatus. Scale bar 20μm. (**c**) Analysis of single-strand breaks by denaturing agarose gel electrophoresis of DNA isolated from~8000 *H. exemplaris* at indicated time points post-irradiation (100Gy or 1000Gy γ-rays from $^{137}$Cs source). (-) indicates DNA from control, non-irradiated tardigrades collected and processed in parallel to treated samples from indicated time points. MW corresponds to the molecular weight ladder. * indicates a discrete band of single-stranded DNA detected in *H. exemplaris* genomic DNA. Arrowhead indicates high molecular weight single-stranded DNA that is not resolved by agarose gel electrophoresis. (-) lanes show DNA from control tardigrades processed in parallel to irradiated tardigrades at 4hr or 8h30 post-irradiation as indicated.

The online version of this article includes the following source data and figure supplement(s) for figure 1:

**Source data 1.** Zip file containing all the raw 16 bit images used in *Figure 1a*.

**Source data 2.** Pdf file showing annotated uncropped images used in *Figure 1a*.

**Source data 3.** Raw 16 bit image used in *Figure 1c*.

**Figure supplement 1.** Characterization of anti-phospho-H2AX antibody.

**Figure supplement 2.** Analysis of DNA damage after γ-ray irradiation.

**Figure supplement 3.** DNA damage and synthesis after 1000Gy γ-ray irradiation.

**Figure supplement 3—source data 1.** Raw 16 bit image used in *Figure 1—figure supplement 3b*.

detected in control animals were irreversibly damaged by the 1000 Gy irradiation (*Figure 1—figure supplement 3a*). Together, these results demonstrate the dose-dependent induction and repair of DSBs in response to IR. Phospho-H2AX immunolabeling experiments also suggested that 1000 Gy induces irreversible damage in the gonads and dividing intestinal cells.

Next, we assessed the physical integrity of genomic DNA (gDNA) at several time points after irradiation. Samples from *Figure 1a* were run in native agarose gels and irradiated samples were found to be indistinguishable from non-irradiated controls (*Figure 1—figure supplement 3b*), showing DSBs and the resulting DNA fragmentation could not be detected in this experimental setting. SSBs were evaluated by migrating DNA samples in denaturing agarose gels (*Figure 1c*). DNA from control, untreated tardigrades appeared as a predominant band running above the 20 kb marker with a smear. The smear, likely due to the harsh extraction conditions needed for tardigrade cuticle lysis, extended down between 20 kb and 10 kb markers where a discrete band, of unknown origin, could be detected (*Figure 1c*). At 30 min after 1000 Gy irradiation, intensity of the high molecular weight band was drastically reduced, and DNA detected in the smear between 10 kb and 20 kb was strongly increased. In addition, the discrete band could no longer be detected. This clearly indicates that 1000 Gy IR induces high rates of SSBs. Considering that the majority of DNA fragments detected had a size of 10–20 kb and that the discrete band of 10–20 kb could no longer be detected, we can roughly evaluate that there is approximately 1 SSB every 10–20 kb. This corresponds to induction of SSBs at a rate of 0.05–0.1 SSB/Mb/Gy. Between 4 hr and 24 hr, the DNA migration profile was progressively restored and between 24 hr and 73 hr, it was identical to controls. Similar results were observed with the 100 Gy dose (*Figure 1c*). However, compared to 1000 Gy, the changes observed were not as marked and the discrete 10–20 kb band could always be detected, indicating SSBs were induced at lower rates. These results indicate that SSBs are inflicted by IR in a dose-dependent manner, roughly estimated to 0.05–0.1 SSB/Mb/Gy, and progressively repaired within the next 24–73 hr (*Figure 1c*).

## *H. exemplaris* strongly overexpresses canonical DNA repair genes as well as RNF146 and TDR1, a novel tardigrade-specific gene, in response to IR

To examine the gene expression changes associated with tardigrade response to IR, we performed RNA sequencing of *H. exemplaris* collected 4 hr after irradiation. The analysis revealed that 421 genes were overexpressed more than 4-fold (with an adjusted p-value<0.05) including 120 overexpressed

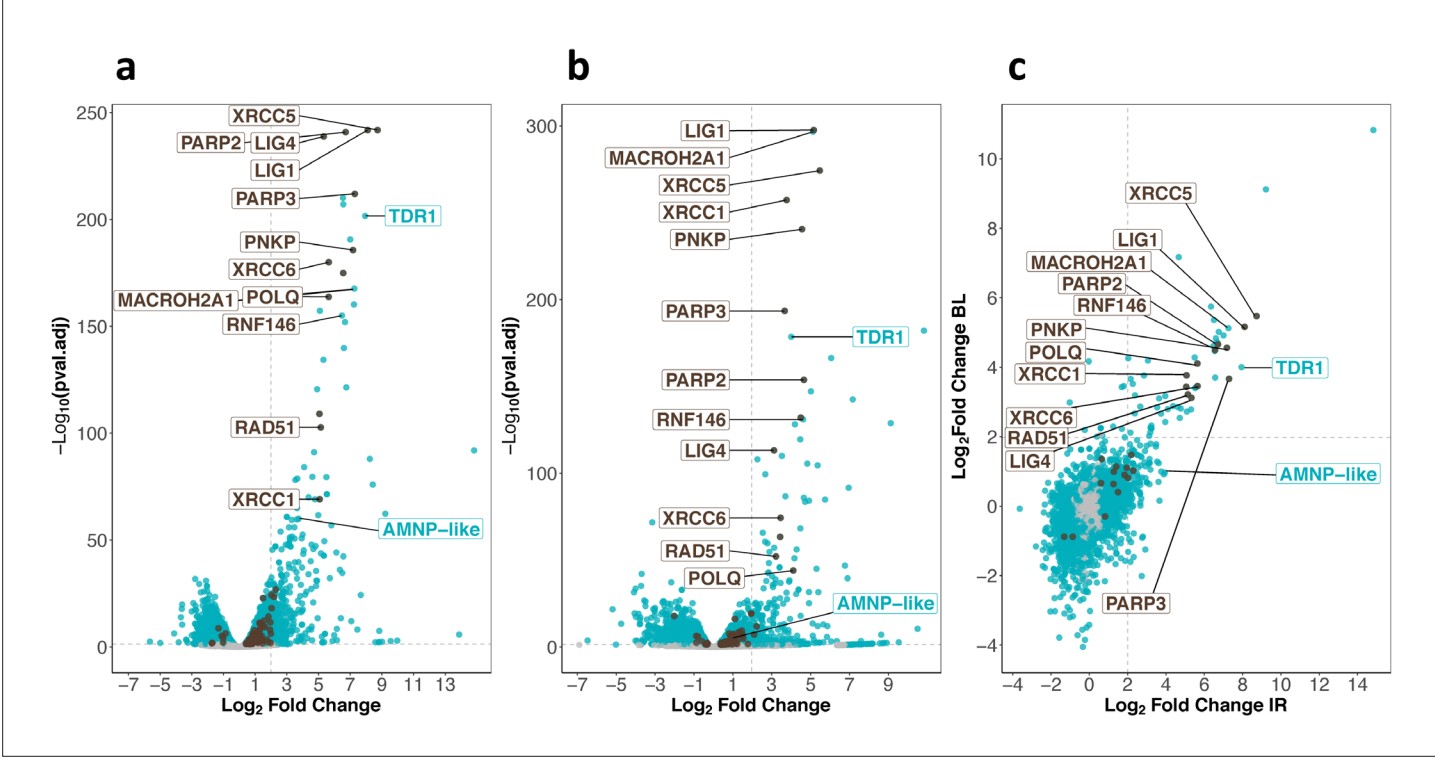

**Figure 2.** Transcriptomic response of *H. exemplaris* to ionizing radiation (IR) and Bleomycin. (**a**) and (**b**) Volcano plots representing Log$_2$ Fold Change and adjusted p-value (−log base 10) of RNA levels between *H. exemplaris* irradiated with 1000Gy γ-rays and untreated controls (n=3) (**a**) and between *H. exemplaris* treated with 100μM Bleomycin for 4days and untreated controls (n=3) (**b**). The vertical dotted lines indicate the Log$_2$ Fold Change value of 2 (Fold Change of 4). (**c**) Correlation between Log$_2$ Fold Change after exposure to IR and after Bleomycin (BL) treatment for abundant transcripts (with baseMean>500 after DESeq2 analysis). The horizontal and vertical dotted lines indicate the Log$_2$ Fold Change value of 2 (Fold Change of 4). Blue dots represent transcripts with a Log$_2$ Fold Change with an adjusted p-value<0.05. Brown dots indicate transcripts of DNA repair genes (based on KEGG DNA repair and recombination gene group ko03400) that have a Log$_2$ Fold Change with adjusted p-value<0.05. Gray dots represent transcripts with a Log$_2$ Fold Change with an adjusted p-value>0.05. Brown labels indicate representative strongly upregulated genes of DNA repair. Blue labels indicate two tardigrade-specific genes induced in response to IR: the TDR1 gene identified in this work, and the AMNP-like gene (BV898_10264), a member of the family of AMNP/g12777-like genes upregulated in response to desiccation and UVC (*Yoshida et al., 2022*).

The online version of this article includes the following source data and figure supplement(s) for figure 2:

**Source data 1.** Table of differentially expressed genes after ionizing radiation (IR) in *H. exemplaris*.

**Source data 2.** Table of differentially expressed genes after Bleomycin treatment in *H. exemplaris*.

**Source data 3.** Table of most abundant (baseMean>500) differentially expressed genes after ionizing radiation (IR) and Bleomycin treatment in *H. exemplaris*.

**Figure supplement 1.** Abundance of *H. exemplaris* differentially expressed genes after ionizing radiation (IR) and Bleomycin treatment.

**Figure supplement 2.** g:Profiler analysis of differentially expressed genes with adjusted p-value<0.05 in *H. exemplaris* after ionizing radiation (IR).

**Figure supplement 3.** Relative abundance of selected genes represented in *Figure 2a*.

**Figure supplement 4.** Relative abundance of selected genes represented in *Figure 2b*.

more than 16-fold (*Figure 2a*, *Figure 2—source data 1*, *Figure 2—figure supplement 1*). The Gene Ontology analysis of overexpressed genes highlighted a strong enrichment of DNA repair genes (*Figure 2a*, *Figure 2—figure supplement 2*). In particular, genes for both major pathways of DNA DSB repair, HR and NHEJ, were among the most strongly stimulated genes. Examples are genes for RAD51 and MACROH2A1 in HR (*Khurana et al., 2014*; *Baumann and West, 1998*) and XRCC5 and XRCC6 in NHEJ (*Doherty and Jackson, 2001*; *Figure 2*). The gene for POLQ, the key player of the alternative end joining pathway of DNA DSB repair (*Mateos-Gomez et al., 2015*), was also strongly upregulated (*Figure 2a*). Also notable among most strongly overexpressed genes were genes for XRCC1, PNKP, and LIG1 in base excision repair (*Whitehouse et al., 2001*; *Krokan and Bjørås, 2013*), along with genes for PARP2 and PARP3, which catalyze PARylation of many DNA repair proteins

(*Pascal, 2018*) and RNF146 (*Figure 2a*). Interestingly, RNF146 is a ubiquitin ligase that has been reported to be important for tolerance to IR in human cells by targeting PARylated XRCC5, XRCC6, and XRCC1 for degradation (*Kang et al., 2011*). Our results suggest that RNF146 upregulation could contribute to the remarkable resistance of tardigrades to IR.

Among overexpressed genes, we also observed AMNP gene family members (*Yoshida et al., 2022*) (one representative was labeled AMNP-like, *Figure 2a*). AMNP genes encode recently discovered tardigrade-specific Mn-peroxidases which are overexpressed in response to desiccation and UVC in *R. varieornatus* (*Yoshida et al., 2022*). AMNP gene g12777 was shown to increase tolerance to oxidative stress when expressed in human cells (*Yoshida et al., 2022*). Based on our results, it is possible that AMNP genes such as the AMNP-like gene identified here could contribute to resistance to IR by increasing tolerance to the associated oxidative stress.

In parallel, we also determined the transcriptomic response of *H. exemplaris* to Bleomycin, a well-known radiomimetic drug (*Bolzán and Bianchi, 2018*; *Figure 2b*, *Figure 2—source data 2*, *Figure 2—figure supplement 1*). In preliminary experiments, we found that *H. exemplaris* tardigrades survived for several days in the presence of 100 µM Bleomycin, suggesting that *H. exemplaris* could resist chronic genotoxic stress. We hypothesized that key genes of resistance to acute genotoxic stress induced by IR would also be induced by Bleomycin treatment. As expected, the correlation between highly expressed genes after IR and after Bleomycin treatment (with baseMean>500, *Figure 2c* and *Figure 2—source data 3*) was strong for most upregulated DNA repair genes such as XRCC5, XRCC6, PARP2, PARP3, XRCC1, LIG4, LIG1, and RNF146 (*Figure 2c* and *Figure 2—figure supplements 3 and 4*). Importantly, in addition to DNA repair genes, several genes of unknown function were also strongly overexpressed in both conditions and considered as promising candidates for a potential role in resistance to IR. One such gene, which we named TDR1 (for Tardigrade DNA damage Response 1), was chosen for further investigation. Oxford Nanopore Technology (ONT) long read sequencing and cDNA cloning of TDR1 allowed us to determine the predicted TDR1 protein sequence which is 146 amino acids long (*Supplementary files 1 and 2*). We observed that the current genome assembly predicts a partially truncated TDR1 protein sequence, BV898_14257, due to an assembly error (*Supplementary file 1*). Our BLAST analysis against NCBI nucleotide non-redundant database suggested that TDR1 is a novel tardigrade-specific gene as no homolog could be found in any other ecdysozoan (*Supplementary file 3*).

## Analysis of proteomic response to IR in *H. exemplaris* confirms overexpression of TDR1

We next examined whether stimulation of gene expression at the RNA level led to increased protein levels and in particular, whether TDR1 protein was indeed overexpressed. For this purpose, we first generated specific antibodies to *H. exemplaris* TDR1, XRCC5, XRCC6, and Dsup proteins. Protein extracts from *H. exemplaris* treated with Bleomycin for 4 days or 1000 Gy of γ-rays at 4 hr and 24 hr post-irradiation were compared to untreated controls. The apparent molecular weight of the TDR1 protein detected on western blots was consistent with the expected 16 kD predicted from the 146 amino acid long sequence (*Figure 3a*). Remarkably, similar to phospho-H2AX, TDR1 was only detected after the induction of DNA damage (*Figure 3a*). XRCC5 and XRCC6 protein levels were also stimulated by both Bleomycin and IR treatments, although the fold stimulation was much lower than at the RNA level (*Figure 3a*, *Figure 3—figure supplement 1*). Furthermore, we checked expression of *He*-Dsup homolog in *H. exemplaris* (*Chavez et al., 2019*), which remained constant at the RNA level (see BV898_01301, *Figure 2—source data 1 and 2*), and found that it also remained stable at the protein level after the induction of DNA damage (*Figure 3a*).

To ensure that the observed stimulation was due to new protein synthesis, we treated tardigrades with the translation inhibitor cycloheximide before irradiation (*Figure 3—figure supplement 2*). As expected, no increase in TDR1, XRCC5, or XRCC6 protein levels could be detected after irradiation in extracts from animals treated with cycloheximide (*Figure 3—figure supplement 2b and c*). In particular, TDR1 protein could not be detected when animals were treated with cycloheximide, further confirming that TDR1 is strongly overexpressed in response to IR.

To further extend the analysis of the protein-level response to IR, we conducted an unbiased proteome analysis of *H. exemplaris* at 4 hr and 24 hr after irradiation and after Bleomycin treatment using mass spectrometry-based quantitative proteomics. More than 5600 proteins could be detected

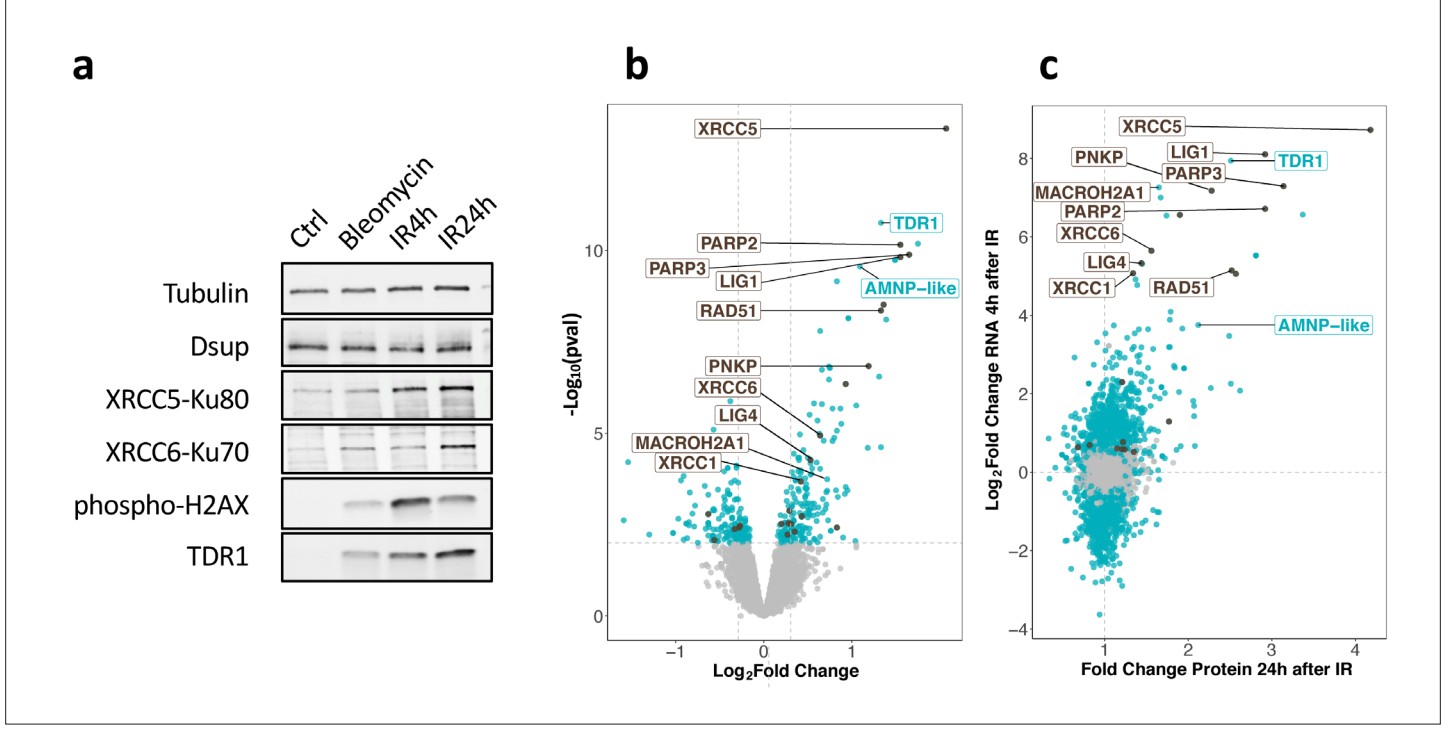

**Figure 3.** Changes in protein expression in *H. exemplaris* after exposure to ionizing radiation (IR). (**a**) Western blot analysis of *He*-TDR1, *He*-XRCC5, *He*-XRCC6 (among the most strongly stimulated genes at the RNA level) and *He*-Dsup (not stimulated at the RNA level) in irradiated *H. exemplaris* tardigrades control, untreated *H. exemplaris* (Ctrl) or *H. exemplaris* treated with 100µM Bleomycin for 4days, or with 1000Gy γ-rays and extracts prepared at indicated times post-irradiation (IR4h and IR24h). Alpha-tubulin was used for normalization and phospho-H2AX for showing induction of DNA double-strand breaks. Quantification of four independent experiments are shown in *Figure 3—figure supplement 1c*. Molecular weight marker present in uncropped western blots (*Figure 3—source data 2*) is consistent with the expected 16kDa size of TDR1. (**b**) Volcano plot representing $\text{Log}_2$ Fold Change and $-\text{Log}_{10}$(limma p-value) of proteins between *H. exemplaris* 24hr post-irradiation with 1000Gy γ-rays and untreated control animals (n=4). Blue dots represent proteins with a $\text{Log}_2$ Fold Change with a $-\text{Log}_{10}$(limma p-value)≥2. Brown dots represent DNA repair proteins (based on KEGG DNA repair and recombination gene group ko03400) with $-\text{Log}_{10}$(limma p-value)≥2. Gray points represent proteins with $\text{Log}_2$ Fold Change with $-\text{Log}10$(limma p-value)<2and the vertical gray lines delimit $\text{Log}_2$ Fold Change>0.3or <−0.3. Brown labels indicate representative strongly upregulated genes of DNA repair. Blue labels indicate two tardigrade-specific genes induced in response to IR: the TDR1 gene identified in this work, and the AMNP-like gene (BV898_10264), a member of the family of AMNP/g12777-like genes upregulated in response to desiccation and UVC (*Yoshida et al., 2022*). (**c**) Correlation between Fold Changes of protein levels 24hr post-irradiation with 1000Gy (as measured in (**b**)) and $\text{Log}_2$ Fold Change of RNA levels 4hr post-irradiation (as measured in *Figure 2a*).

The online version of this article includes the following source data and figure supplement(s) for figure 3:

**Source data 1.** Zip file containing all the 16 bit images used in *Figure 3a*.

**Source data 2.** Pdf file showing annotated uncropped images used in *Figure 3a*.

**Source data 3.** Table of differentially expressed proteins after ionizing radiation (IR) 4hr or 24hr post-irradiation and after Bleomycin treatment 5days in *H. exemplaris*.

**Figure supplement 1.** Expression of selected proteins by western blot and quantifications.

**Figure supplement 1—source data 1.** Zip file containing all the raw 16 bit images used in *Figure 3—figure supplement 1a*.

**Figure supplement 1—source data 2.** Pdf file showing the annotated uncropped images used in *Figure 3—figure supplement 1a*.

**Figure supplement 2.** Impact of cycloheximide on protein levels in *H. exemplaris* after exposure to ionizing radiation (IR).

**Figure supplement 2—source data 1.** Zip file containing all the raw 16 bit images used in *Figure 3—figure supplement 2b*.

**Figure supplement 2—source data 2.** Pdf file showing the annotated uncropped images used in *Figure 3—figure supplement 2b*.

in all conditions (*Figure 3—source data 1*). Among them, 58, 266, and 185 proteins were found to be differentially abundant at 4 hr post-irradiation, 24 hr post-irradiation, and after Bleomycin treatment, respectively compared to control tardigrades ($\text{Log}_2$ Fold Change>0.3 and limma p-value<0.01, leading to a Benjamini-Hochberg FDR<3%, *Figure 3b*, *Table 1*, and *Figure 3—source data 1*). We observed a good correlation between stimulation at RNA and protein levels (*Figure 3*). It is worth

**Table 1.** Proteomic analysis metrics: numbers of differentially expressed (DE) proteins (with limma p-value<0.01and Log$_2$ Fold Change<−0.3 or >0.3) for each indicated condition in *H. exemplaris*. The numbers of tardigrade-specific DE proteins are also indicated. Nine tardigrade-specific DE proteins were common to the three conditions, the corresponding list is provided in *Supplementary file 4*. Tardigrade-specific proteins are defined as detailed in the Materials and methods section. IR4h, 4hr post-1000 Gy $\gamma$-ray irradiation; IR24h, 24hr post-1000 Gy $\gamma$-ray irradiation; Ctrl, Control.

| | IR4h *vs* Ctrl | IR24h *vs* Ctrl | Bleomycin vs Ctrl |
|---|---|---|---|
| *Total number of proteins identified in the three conditions* | 5625 | | |
| **DE proteins** | **58** | **266** | **185** |
| DE proteins up | 42 | 168 | 128 |
| DE proteins down | 16 | 98 | 57 |
| *DE proteins in the three conditions* | 36 | | |
| **Tardigrade-specific DE proteins** | **13** | **61** | **70** |
| Tardigrade-specific DE proteins up | 11 | 52 | 47 |
| Tardigrade-specific DE proteins down | 2 | 9 | 23 |
| *Tardigrade-specific DE proteins in the three conditions* | 9 | | |

noting that the fold changes observed for proteins were smaller than those obtained for mRNAs, possibly due to the use of an isobaric multiplexed quantitative proteomic strategy known to compress ratios (*Hogrebe et al., 2018*). For strongly overexpressed canonical DNA repair genes discussed above, we confirmed significantly increased protein levels in response to IR (*Figure 3b*). RNF146, in contrast, could not be detected, likely due to limited sensitivity of our mass spectrometry-based quantitative proteomics. Importantly, despite the small size of the predicted TDR1 protein, we detected four different TDR1-related peptides, providing direct evidence of strong TDR1 overexpression in response to IR (*Figure 3—source data 1*).

## Conservation of TDR1 and transcriptional response to IR in other tardigrade species

To gain insight into the importance of the upregulation of TDR1 and DNA repair genes in resistance to IR, we chose to investigate its conservation in other tardigrade species. We successfully reared two other species in the lab: *A. antarcticus,* from the Hypsibioidea superfamily, known for its resistance to high doses of UV, likely related to its exposure to high levels of UV in its natural Antarctic habitat (*Giovannini et al., 2018*), and *P. fairbanksi* (*Guidetti et al., 2019*), which was reared from a garden moss and was of high interest as a representative of Macrobiotoidea, a major tardigrade superfamily considered to have diverged from Hypsibioidea more than 250 My ago (*Regier et al., 2004*). It was in *Paramacrobiotus areolatus*, which also belongs to Macrobiotoidea, that the first demonstration of

**Table 2.** Number of differentially expressed genes (DEG with adjusted p-value<0.05) after ionizing radiation (IR) with 1000Gy $\gamma$-rays *vs* untreated in three species (*H. exemplaris, A. antarcticus, P. fairbanksi*) and Bleomycin treatment for 4 or 5days in *H. exemplaris* and *A. antarcticus*.
A heatmap of the 50 putative orthologous upregulated genes common to all conditions is given in *Figure 4—figure supplement 5*.

| γ-irradiation *vs* control | *H. exemplaris* | *A. antarcticus* | *P. fairbanksi* | Bleomycin vs control | *H. exemplaris* | *A. antarcticus* |
|---|---|---|---|---|---|---|
| Total number of DEG | 6209 | 3708 | 7515 | Total number of DEG | 5116 | 1458 |
| DEG up | 3178 | 1875 | 3687 | DEG up | 2284 | 399 |
| DEG down | 3031 | 1833 | 3828 | DEG down | 1113 | 1059 |

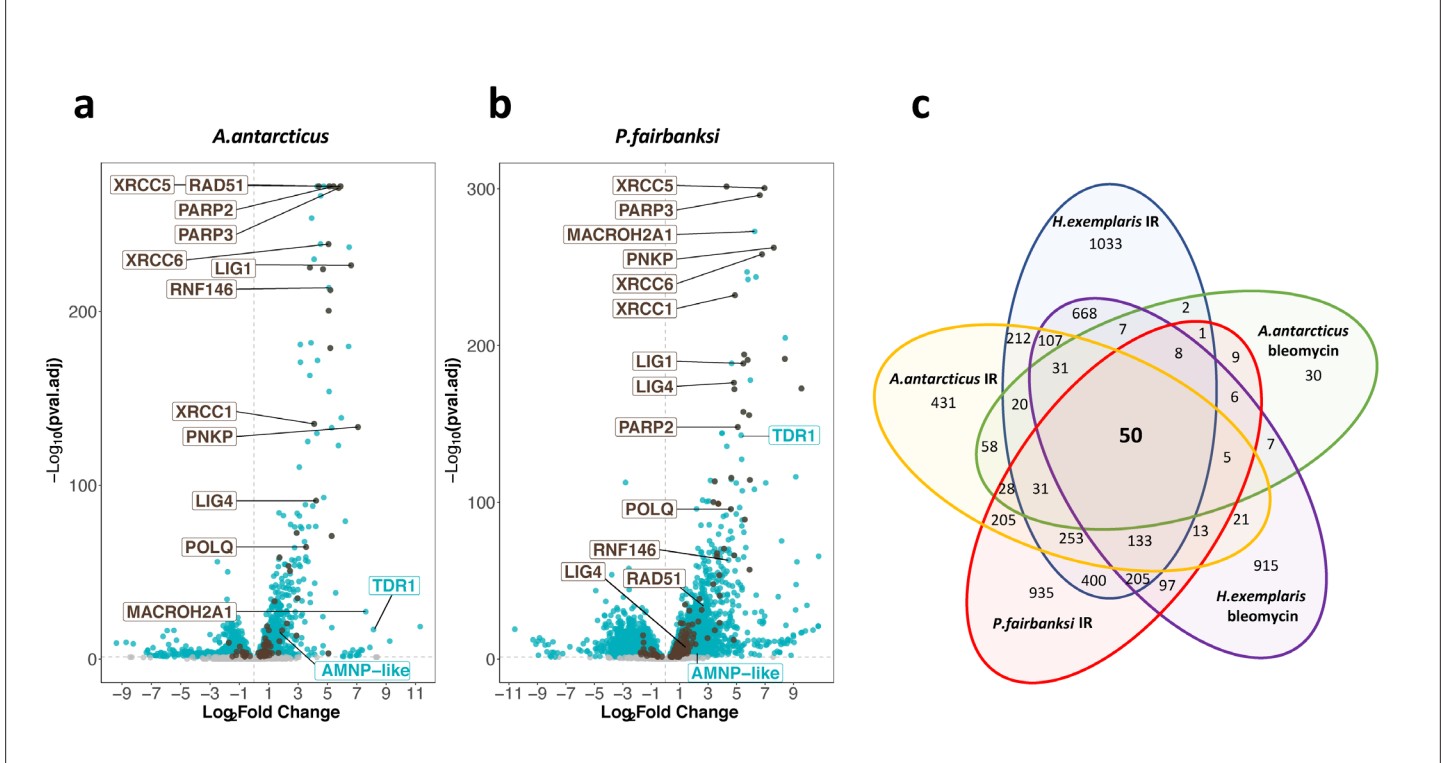

**Figure 4.** Transcriptomic response of *A. antarcticus* and *P. fairbanksi* to ionizing radiation (IR). (**a**) and (**b**) volcano plots representing Log$_2$ Fold Change and adjusted p-value ($-$log base 10) of RNA levels after irradiation with 1000Gy γ-rays between irradiated *A. antarcticus* and untreated controls (n=3) (**a**) and between irradiated *P. fairbanksi* and untreated controls (n=3) (**b**). Blue dots represent transcripts with an adjusted p-value<0.05. Brown dots indicate transcripts of DNA repair genes (based on KEGG DNA repair and recombination gene group ko03400) with an adjusted p-value<0.05. Brown labels indicate representative strongly upregulated genes of DNA repair. Blue labels indicate two tardigrade-specific genes induced in response to IR: the TDR1 gene identified in this work, and the AMNP-like gene (BV898_10264), a member of the family of AMNP/g12777-like genes upregulated in response to desiccation and UVC (*Yoshida et al., 2021*). (**c**) Venn diagram showing upregulated genes with an adjusted p-value<0.05 common to the transcriptomic response to IR in the three species analyzed and to Bleomycin in *H. exemplaris* and *A. antarcticus*.

The online version of this article includes the following source data and figure supplement(s) for figure 4:

**Source data 1.** Table of differentially expressed genes after ionizing radiation (IR) in *A. antarcticus*.

**Source data 2.** Table of differentially expressed genes after Bleomycin treatment in *A. antarcticus*.

**Source data 3.** Table of differentially expressed genes after ionizing radiation (IR) in *P. fairbanksi*.

**Figure supplement 1.** Abundance of differentially expressed genes of *A. antarcticus* and *P. fairbanksi* after ionizing radiation (IR) and of *A. antarcticus* after Bleomycin treatment.

**Figure supplement 2.** Relative abundance of selected genes represented in *Figure 4a* and *Figure 4—figure supplement 1a*.

**Figure supplement 3.** Relative abundance of selected genes represented in *Figure 4—figure supplement 1b*.

**Figure supplement 4.** Relative abundance of selected genes represented in *Figure 4b* and *Figure 4—figure supplement 1c*.

**Figure supplement 5.** Heatmap of 50 putative orthologous genes upregulated in response to ionizing radiation (IR) in all the three species analyzed, *H. exemplaris*, *A. antarcticus*, and *P. fairbanksi*, and in response to Bleomycin in both *H. exemplaris* and *A. antarcticus*.

resistance to IR was carried out (with an LD50$_{24hr}$ of 5700 Gy) (*May, 1964*). Importantly, species of Macrobiotoidea examined so far lack Dsup homologs (*Arakawa, 2022*).

We found by visual inspection of animals after IR that *A. antarcticus* and *P. fairbanksi* readily survived exposure to 1000 Gy. As done above in *H. exemplaris*, we therefore examined genes differentially expressed 4 hr after 1000 Gy IR. In both species, we found numerous genes to be significantly overexpressed in response to IR, and similar to what we observed in *H. exemplaris*, upregulation was often remarkably strong (*Figure 4a and b*, *Table 2*, *Figure 4—source data 1–3*, *Figure 4—figure supplements 1–4*). Crucially, we identified TDR1 homologs in transcriptomes of *A. antarcticus* and *P. fairbanksi* and just like in *H. exemplaris*, these TDR1 homologs were among the most overexpressed genes after IR in both species and in response to Bleomycin treatment of *A. antarcticus* (*Table 2*,

*Figure 4—source data 1–3*), strongly suggesting a conserved role of TDR1 in resistance to IR. In contrast, as expected from previous studies, we could identify a Dsup homolog in *A. antarcticus* (Aant_geneID_rb_14333, *Figure 4—source data 1 and 2*), from the Hypsibioidea superfamily, but not in *P. fairbanksi* from Macrobiotoidea.

Furthermore, similar to *H. exemplaris*, Gene Ontology analysis of overexpressed genes highlighted a robust enrichment of DNA repair genes in *A. antarcticus* and *P. fairbanksi* in response to IR (*Supplementary file 5a and b*). Notably, a high proportion of genes of the main repair pathways of DNA damages caused by IR (DSB and SSB repair, and base excision repair) were significantly overexpressed after IR in all three species (*Supplementary file 5c and d*) and as in *H. exemplaris*, among the genes with the strongest overexpression in *A. antarcticus* and *P. fairbanksi,* we observed the canonical DNA repair genes for XRCC5, XRCC6, XRCC1, PARP2, PARP3, as well as the gene for RNF146. Interestingly, a set of 50 putative orthologous genes was upregulated in response to IR in all three species, suggesting a conserved signaling and transcriptional program is involved in response to IR between the distantly related Hypsibioidea and Macrobiotoidea superfamilies (*Figure 4—figure supplement 5*).

## *He*-TDR1 interacts directly with DNA in vitro and co-localizes with DNA in transgenic tardigrades

In addition to the three species studied, BLAST searches against recent tardigrade transcriptomes enabled the identification of potential TDR1 homologs in other tardigrade species, which all belong to the Macrobiotoidea superfamily (*Figure 5a*, *Supplementary file 2*). The TDR1 proteins are predicted to be 146–291 amino acids long, with the C-terminal part showing the highest similarity (*Figure 5a*). Interestingly, TDR1 proteins contain a relatively high proportion of basic amino acid residues (20.5% of K or R amino acids for TDR1 of *H. exemplaris, He*-TDR1), including at conserved positions in the C-terminal domain (*Figure 5a*). This led us to wonder if TDR1 might interact directly with DNA. To investigate this possibility, we purified recombinant *He*-TDR1 (*Figure 5—figure supplement 1*) and tested its interaction with DNA using gel shift assays. As shown in *Figure 5b and c*, when circular or linear plasmid DNA was incubated with increasing concentrations of *He*-TDR1, a shift in plasmid mobility was detected in agarose gel electrophoresis, indicating the formation of a complex between *He*-TDR1 and DNA. The observed binding of *He*-TDR1 at a ratio of 1 He-TDR1 protein to every 3 bp of DNA is similar to the binding reported for non-sequence-specific DNA-binding proteins such as the Rad51 recombinase (*Zaitseva et al., 1999*). Upon adding the highest amounts of *He*-TDR1, we noted that the amount of plasmid DNA detected by ethidium bromide staining appeared to decrease. We ruled out that plasmid DNA was degraded during incubation by performing proteinase K treatment which revealed that the amounts of intact plasmid DNA had not changed after incubation with *He*-TDR1. As an alternative explanation, we considered that at high *He*-TDR1 concentrations, *He*-TDR1 and DNA might form aggregates that could not enter the gel. To explore this possibility, we examined mixes of *He*-TDR1-GFP and plasmid DNA by fluorescence microscopy. At ratios at which complex formation was detected by agarose gel electrophoresis (*Figure 5b and c*), we observed fluorescent spots in the samples, suggesting the presence of large protein-DNA aggregates (of 2–5 μm) likely unable to enter the agarose gels (*Figure 5—figure supplement 2*).

To further examine the potential interaction of *He*-TDR1 with DNA in vivo, we generated a tardigrade expression plasmid with *He*-TDR1-mNeonGreen cDNA downstream of *He-Actin* promoter sequences and introduced it into tardigrade cells using a recently reported protocol (*Tanaka et al., 2023*). *He*-TDR1-mNeonGreen was easily detected in muscle cells, likely due to high muscle-specific activity of the *He-Actin* promoter, and predominantly localized to nuclei, as observed by confocal microscopy (*Figure 5d and e*). Importantly, *He*-TDR1 co-localized with Hoechst staining, suggesting *He*-TDR1 is able to interact with DNA in vivo. In summary, these experiments clearly documented interaction of *He*-TDR1 with DNA but also revealed its unexpected ability to compact DNA into aggregates.

## Expression of TDR1 proteins diminishes the number of phospho-H2AX foci in human U2OS cells treated with Bleomycin

Next, we aimed to investigate whether the expression of TDR1 could impact the number of phospho-H2AX foci detected upon treatment of human U2OS cells with the radiomimetic drug Bleomycin. When

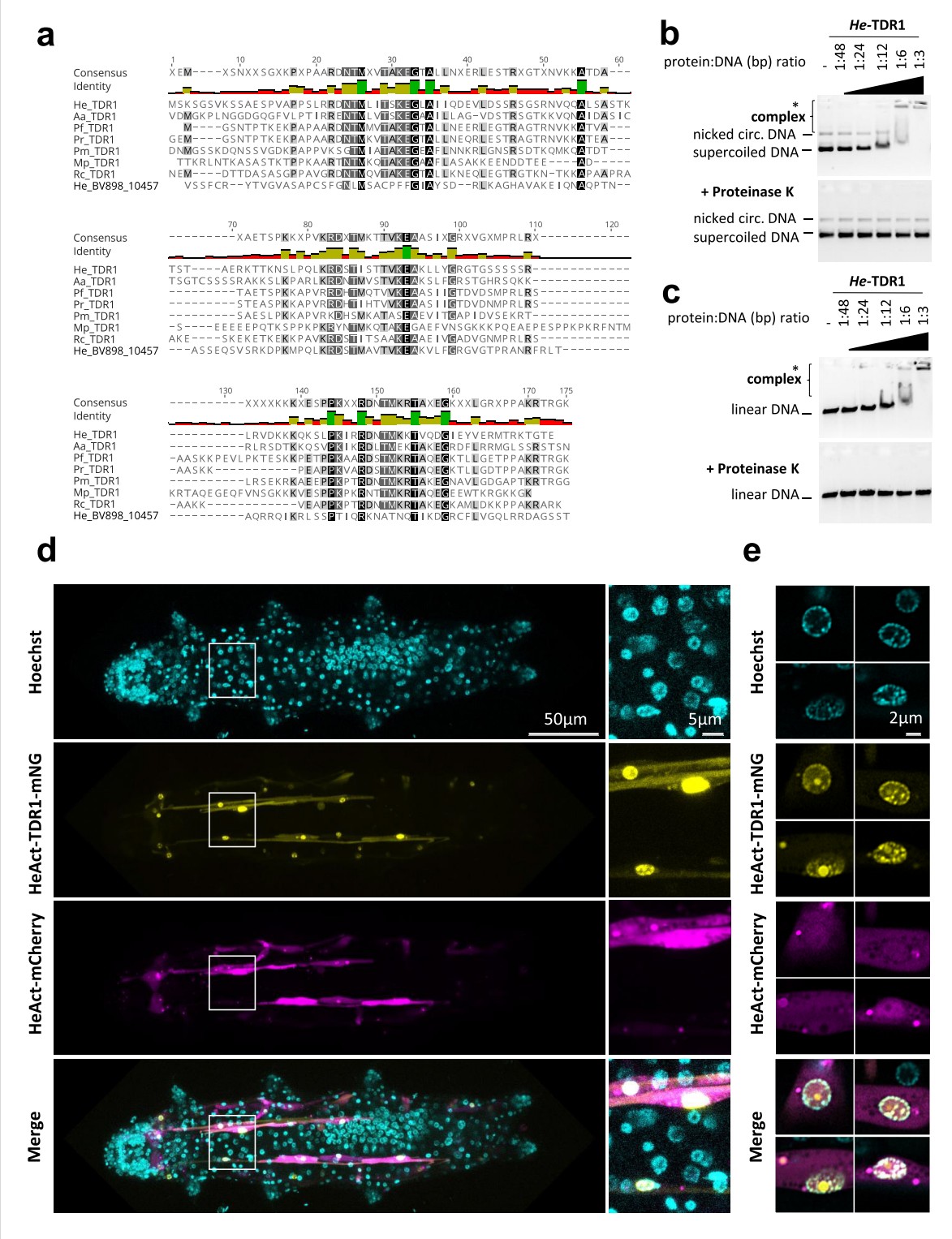

**Figure 5.** *He*-TDR1 interacts directly with DNA. (**a**) Sequence alignment of the conserved C-terminal domain of TDR1 proteins from *H. exemplaris* (He), *A. antarcticus* (Aa), *P. fairbanksi* (Pf) (identified in this work), and from *Paramacrobiotus richtersi* (Pr) (NCBI transcriptome assembly GFGY00000000.1, now known as *Paramacrobiotus spatialis*), *P. metropolitanus* (Pm), *Richtersius coronifer* (Rc) (**Kamilari et al., 2019**), *Mesobiotus philippinicus* (Mp) (**Mapalo et al., 2020**). He_BV898_10457 corresponds to a paralog of *He*TDR1 in *H. exemplaris* with weaker sequence identity to *He*-TDR1 than TDR1 homologs from other species. (**b–c**) Gel shift assay of recombinant *He*-TDR1 with circular plasmid (**b**) or linear plasmid (**c**). Mixes of plasmid DNA and recombinant *He*-TDR1 at indicated protein to DNA (bp) ratios were incubated at 25°C for 20min and migrated, either directly or after proteinase K digestion, at

*Figure 5 continued on next page*

*Figure 5 continued*

room temperature on 0.75% agarose with ethidium bromide. Fluorescence was revealed with a ChemiDoc MP imager. Complexes of plasmid DNA and recombinant *He*-TDR1 are indicated by a bracket. High molecular weight complexes that remained in the loading wells and did not migrate into the gel are indicated by an asterisk. (**d**) Expression of *He*-TDR1-mNeonGreen in transient transgenic *H. exemplaris* tardigrades. Expression plasmids of *He*-TDR1-mNeonGreen (mNG) and mCherry (both under control of the *He*-Actin promoter) were microinjected into the body fluid of *H. exemplaris* adults and electroporation was performed to induce delivery into cells following the protocol of *Tanaka et al., 2023*. Confocal microscopy was carried out on live animals immobilized in carbonated water at day 8 post-microinjection after 2days of treatment with 20μM Hoechst 33342 to stain nuclei. Maximum projections of confocal Z-stack are shown. (**e**) High-resolution imaging of nuclei expressing *He*-TDR1-mNG and Hoechst staining of the nucleus using the Airyscan2 module (one Z-slice is shown). Nuclear *He*-TDR1-mNG is co-localized with Hoechst staining except for one big foci which was observed in some high-resolution images (yellow channel), likely corresponding to nucleolar accumulation of overexpressed *He*-TDR1-mNG.

The online version of this article includes the following source data and figure supplement(s) for figure 5:

**Source data 1.** Zip file containing all the raw 16 bit images used in *Figure 5b and c*.

**Source data 2.** Pdf file showing the annotated uncropped images used in in *Figure 5b and c*.

**Figure supplement 1.** Production of recombinant *He*-TDR1 and *He*-TDR1-GFP.

**Figure supplement 2.** Formation of aggregates of *He*-TDR1 and DNA.

DSBs occur, H2AX is phosphorylated along extended DNA regions near the break and phospho-H2AX foci can be easily detected by immunolabeling, providing a means to indirectly visualize and quantify DSBs in nuclei (*Lowndes and Toh, 2005*). We designed plasmids for expression of TDR1 proteins from different tardigrade species fused to GFP and transfected them into human U2OS cells. After 48 hr, we treated cells with 10 μg/mL Bleomycin to induce DSBs. This allowed us to quantify phospho-H2AX foci in response to Bleomycin by immunolabeling with anti-human phospho-H2AX antibody. As controls, we transfected plasmids expressing either GFP, *Rv*Dsup-GFP, or *He*RNF146-GFP. The quantification of phospho-H2AX was carried out in transfected cells (*Figure 6a* and *Figure 6—figure supplement 1*). As previously demonstrated for *Rv*Dsup (*Hashimoto et al., 2016*) and as expected from the characterization of human RNF146 (*Kang et al., 2011*), expression of *Rv*Dsup-GFP and *He*RNF146-GFP respectively reduced the number of phospho-H2AX foci. This result strongly suggests that *He*RNF146 is a homolog of human RNF146. Moreover, expression of TDR1-GFP fusion proteins from all species tested also significantly reduced the number of phospho-H2AX foci in human cells treated with Bleomycin, supporting the potential role of TDR1 proteins in tardigrade resistance to IR. *Figure 6b* shows that *He*-TDR1-GFP protein was localized in the nucleus of transfected cells, which is consistent with its ability to directly interact with DNA and its nuclear localization after transgenic expression in *H. exemplaris*.

## Discussion

Our study aimed to understand the role of DNA repair in the remarkable radio-resistance of tardigrades. We examined the DNA damage and repair mechanisms in the tardigrade species *H. exemplaris* after exposure to ionizing radiation (IR) and performed comparative transcriptomics in three species of the Tardigrada phylum. Our results indicate that DNA repair plays a major role in the radio-resistance of tardigrades and identified the gene for TDR1, a novel DNA-binding protein highly upregulated in response to IR and likely to play an original function in DNA repair.

### DNA repair plays a major role in resistance of tardigrades to IR

Using an antibody raised against phosphorylated *He*-H2AX, we could detect DSBs by western blot and by immunolabeling (*Figure 1*). Our analysis documented dose-dependent DNA damage and repair taking place after exposure to IR. DNA damage could be detected in virtually all nuclei by immunolabeling. However, at 1000 Gy, phosho-H2AX labeling persisted longer than at 100 Gy in the gonad. Additionally, at 1000 Gy, cell divisions could no longer be detected in the midgut of the digestive system. These two consequences of exposure to higher doses of IR may be due to higher sensitivity of replicating cells to IR and explain why *H. exemplaris* tardigrades no longer lay eggs and become sterile after irradiation with 1000 Gy (*Beltrán-Pardo et al., 2015*).

Using standard agarose gel electrophoresis, we were able to observe that SSBs were induced every 10–20 kb in *H. exemplaris* after exposure to 1000 Gy of γ-rays, indicating a rate of 0.05–0.1 SSB/Gy/Mb (*Figure 1c*). Remarkably, this rate is roughly similar to that reported for cultured human cells which

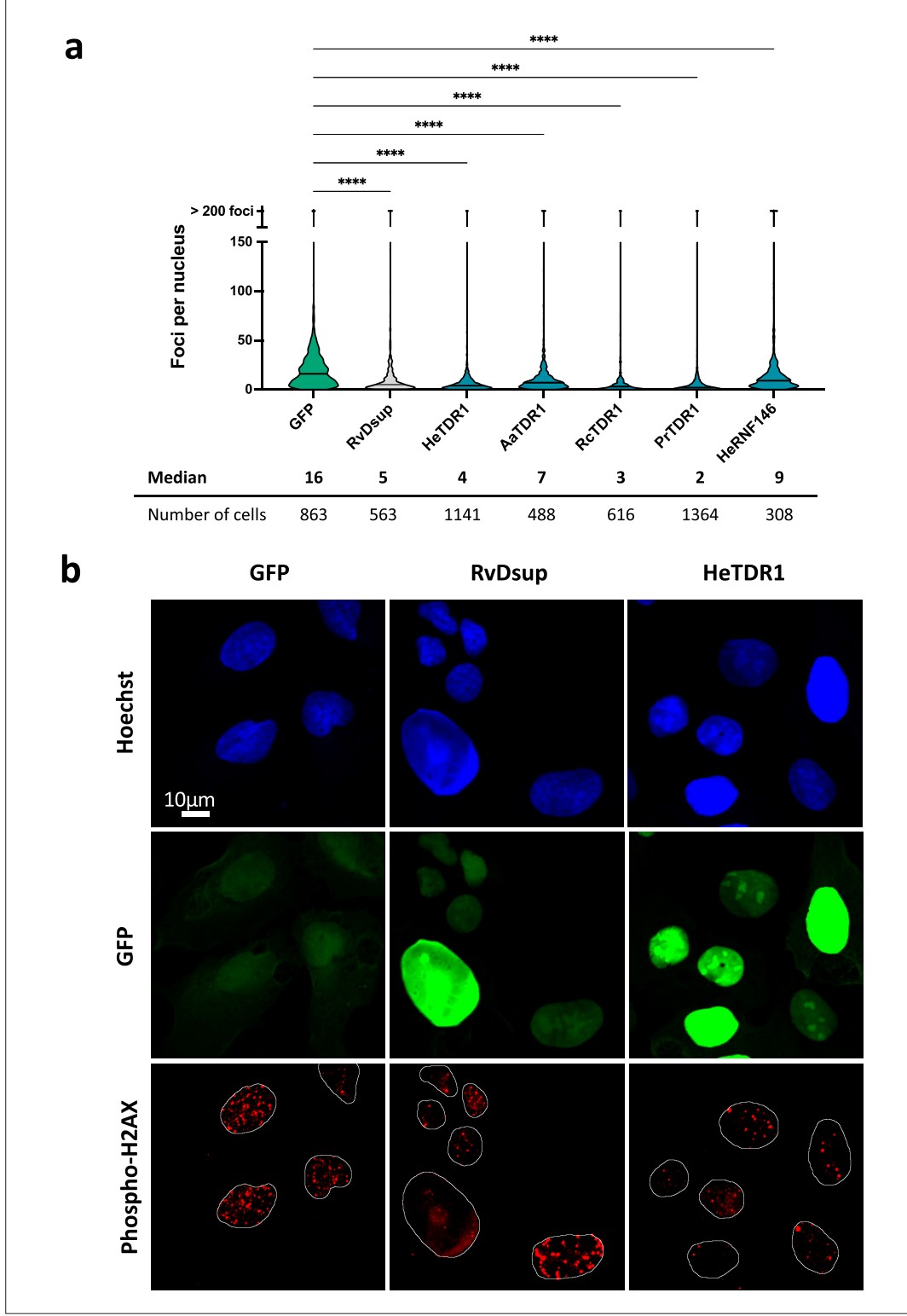

**Figure 6.** Reduced numbers of phospho-H2AX foci after Bleomycin treatment in human U2OS cells expressing TDR1-GFP from multiple tardigrade species. (**a**) Violin plot of the number of phospho-H2AX foci per nucleus of cells expressing the indicated protein. Phospho-H2AX foci were counted after 1 µg/mL Bleomycin 1 hr treatment of U2OS cells electroporated with a plasmid expressing either eGFP (control), RvDsup-GFP, TDR1-GFP from *H. exemplaris* (He), *A. antarcticus* (Aa), *R. coronifer* (Rc), and *P. richtersi* (Pr), He-RNF146-GFP. Cells were fixed with a 4% PFA PBS solution for 1hr, immunolabeled with chicken anti-GFP and mouse anti-phospho-H2AX antibodies

*Figure 6 continued on next page*

*Figure 6 continued*

and imaged by confocal microscopy. **** indicates p<0.0001 (Kruskal-Wallis test). A minimum of 308 nuclei were counted in each experimental condition (n=3). A representative experiment is shown here. Data from independent replicates are given in *Figure 6—figure supplement 1*. (**b**) Representative confocal fluorescence imaging of experiment analyzed in (**a**).Images were taken with identical settings and printed with same thresholding so that signal intensity could be compared. Scale bar corresponds to 10µM.

The online version of this article includes the following figure supplement(s) for figure 6:

**Figure supplement 1.** Independent replicates of experiments of *Figure 6*.

is 0.17 SSB/Gy/Mb (*Mohsin Ali et al., 2004*), showing that high levels of DNA damage are induced after high doses of IR and thus supporting the importance of DNA repair in the radio-resistance of *H. exemplaris* compared to human cells. In radio-resistant rotifers, the rate of DSBs was comparable to non-radio-resistant organisms (*Gladyshev and Meselson, 2008*), also suggesting the importance of DNA repair in radio-resistance. Concerning the role of DNA protection in radio-resistance, further studies would be necessary; in particular, determining the rate of DNA DSBs and testing the importance of Dsup in live *H. exemplaris*. Quantification of phospho-H2AX foci is frequently used as a proxy but given the small size of tardigrade nuclei, standard imaging by confocal microscopy was not sufficient to clearly identify and quantify independent phospho-H2AX foci. Recent developments in super-resolution microscopy could make it possible to perform such quantification in the future (*Tillberg and Chen, 2019*). Pulse field electrophoresis is a method that would allow to directly examine DNA damage, but it would require to disrupt the cuticle and release DNA without causing damage which would confound the analysis of DNA integrity.

## Fine regulation of scaffolding proteins to cope with high rates of DNA damage

We performed comparative transcriptomics in three different species and uncovered the conserved upregulation of a wide number of DNA repair genes in response to IR (*Figure 4* and *Supplementary file 5*). Remarkably, the strongest upregulations, both at RNA and protein levels, were detected for proteins acting early in DNA repair in the different pathways involved: XRCC5/XRCC6 in NHEJ (*Doherty and Jackson, 2001*), POLQ in micro-homology-mediated end joining (MMEJ), XRCC1 in SSB, and PARP2/PARP3 which act as DNA damage sensors common to all DSB repair pathways (*Pandey and Black, 2021*). These early acting proteins either stabilize DNA ends or provide essential scaffolding for subsequent steps of DNA repair. It is possible that producing higher amounts of such proteins is essential to maintain DNA ends long enough for more limiting components of DNA repair to cope with an exceptionally high number of damages. For XRCC5 and XRCC6, our study established, by two independent methods, proteomics and western blot analyses, that the stimulation at the protein level was much more modest (6- and 20-fold at most, *Figure 3—figure supplement 1*) than at the RNA level (420- and 90-fold, respectively). This finding suggests that the abundance of DNA repair proteins does not simply increase massively to quantitatively match high numbers of DNA damages. Interestingly, in response to IR, the RNF146 ubiquitin ligase was also found to be strongly upregulated. RNF146 was previously shown to interact with PARylated XRCC5 and XRCC6 proteins and to target them for degradation by the ubiquitin-proteasome system (*Kang et al., 2011*). To explain the more modest fold stimulation of XRCC5 and XRCC6 at protein levels compared to its massive increase at mRNA levels, it is therefore tempting to speculate that, XRCC5 and XRCC6 protein levels (and perhaps that of other scaffolding complexes of DNA repair) are regulated by a dynamic balance of synthesis, promoted by increased gene transcription, and degradation, made possible by RNF146 upregulation. Consistent with this hypothesis, we found that, similar to human RNF146 (*Kang et al., 2011*), *He*-RNF146 expression in human cells reduced the number of phospho-H2AX foci detected in response to Bleomycin (*Figure 6*).

Further studies should investigate the molecular mechanisms leading to such marked upregulation of RNA levels of these genes. Functional analysis of promoter sequences in transgenic tardigrades is now possible (*Tanaka et al., 2023*) and could help to identify a conserved set of transcription factors and/or co-regulators common to Macrobiotodea and Hypsibioidea tardigrades. Such information would provide original insight into the acquisition of resistance to IR and help analyze its relation to the resistance to desiccation. Another outstanding issue, given the high rates of DNA damage taking

place, is whether DNA repair is accurate. This is particularly relevant for germ line cells where mutations will be transmitted to the progeny and could impact evolution of the species.

## A novel tardigrade-specific DNA-binding protein involved in resistance to IR

Among the genes overexpressed in response to IR in the three species studied, we identified TDR1 as a promising tardigrade-specific candidate (*Figures 3 and 4*). At the functional level, we found that TDR1 protein interacts with DNA (*Figure 5*) and that when expressed in human cells, TDR1 protein can reduce the number of phospho-H2AX foci induced by Bleomycin (*Figure 6*). TDR1 is strongly overexpressed in response to IR (*Figure 3*), suggesting that it favors DNA repair. Proteins directly involved in DNA repair, however, usually accumulate at sites of DNA damage (*Rothkamm et al., 2015*), which did not appear to be the case for TDR1 overexpressed in human cells. Given that TDR1 can form aggregates with DNA in vitro, we speculate that it may favor DNA repair by regulating chromosomal organization. Intriguingly, the DNA-binding activities of TDR1 are reminiscent of DdrC from *D. radiodurans*. DrdC is a small DNA-binding protein which is among the most strongly overexpressed proteins after irradiation of *D. radiodurans* with γ-rays (*Tanaka et al., 2004*) and DdrC forms aggregates with DNA in vitro (*Banneville et al., 2022*; *Bouthier de la Tour et al., 2017*). Further investigations of TDR1 may thus reveal unexpected parallels between mechanisms of DNA repair conferring radio-resistance in tardigrades and bacteria.

Recent progress in tardigrade transgenesis (*Tanaka et al., 2023*) and promising findings of somatic mutagenesis by CRISPR-Cas9 (*Kumagai et al., 2022*) are paving the way toward germ line gene editing in *H. exemplaris*. Knocking out Dsup and TDR1 genes should help to better appreciate their importance in radio-resistance and the underlying mechanisms. The C-terminal portion of TDR1 is conserved in species of Macrobiotoidea and Hypsibioidea superfamilies of the Parachela order of Eutardigrada but absent from *M. inceptum*, the only representative of the Apochela order of Eutardigrada for which transcriptomic data is currently available (*Supplementary file 6*), and from Heterotardigrada. Compared to Dsup, which has only been found in Hypsibioidea, TDR1 appears more widely present and could be a more ancient tardigrade gene. As additional tardigrade species, more fully representing the phylogenetic diversity of the phylum, are reared in laboratory conditions and become amenable to experimental analysis, more novel genes and mechanisms of radio-resistance may become apparent. Generally, evolution of tardigrade-specific gene sequences appears highly dynamic in the phylum (*Arakawa, 2022*), and further sequencing of tardigrade genomes will help get a better picture of gain and loss of tardigrade-specific genes and their relation to resistance to extreme conditions.

In conclusion, our findings suggest that DNA repair is a major contributor to tardigrade radio-resistance. Functional investigations of TDR1, as well as the study of transcriptional regulation in response to IR, will contribute to a deeper understanding of the mechanisms underlying radio-resistance. Additionally, we believe that, as done here, further exploration of tardigrade-specific genes and comparative studies among tardigrade species will shed light on the evolution and diversity of radio-resistance mechanisms in these fascinating organisms.

# Materials and methods
## Tardigrade culture

*H. exemplaris* (strain Z151, Sciento, UK) and *A. antarcticus* (*Giovannini et al., 2018*) were cultivated from one individual in mineral water (Volvic, France) at 16°C with a 14 hr day and 10 hr night cycle and fed with *Chlorella vulgaris* microalgae from Algothèque MNHN, France, or ordered from https://www.ldc-algae.com/ (Greenbloom pure fresh Chlorella). Microalgae were grown in 1× Cyanobacteria BG-11 Freshwater Solution (C3061, Sigma-Aldrich) at 23°C until saturation, collected by centrifugation at 2000×*g* for 5 min and resuspended in Volvic mineral water 10-fold concentrated before feeding. For concentrated large amounts of *H. exemplaris* 0.2% of linseed oil emulsion containing 95% linseed oil (organic), 1% Tween 80 (P4780, Sigma-Aldrich) and 4% (+/−)-α-Tocopherol (T3251, Sigma-Aldrich) could be spread at the bottom of the untreated Petri dishes before adding algae and water. *P. fairbanksi* was isolated from a suburban garden moss by A.D.C. and cultivated in the same conditions adding rotifers isolated from the same moss sample (grown separately in Volvic mineral

water and fed with *C. vulgaris*) of as supplementary food for adults. Identification as *P. fairbanksi* was achieved by morphological and DNA markers (*Supplementary file 7*).

## IR and Bleomycin treatments of tardigrades

Prior to treatments, tardigrades were separated from *Chlorella* by filtering with a cell strainer 70 μm mesh (141379C, Clearline) for *H. exemplaris* or 100 μm mesh (141380C, Clearline) for *A. antarcticus* and *P. fairbanksi*. The cell strainer containing the washed tardigrades was put in a Petri dish containing Volvic mineral water for 3–7 days in order to allow live tardigrades to go out of the cell strainer and obtain tardigrades without any remaining *Chlorella*. Tardigrades were then collected on 40 μm mesh (141378C, Clearline) and washed with Volvic mineral water before proceeding to treatments. For each treatment, tardigrades were collected and split into treated and control samples. Control samples were subjected to the same conditions (temperature, travel [for irradiation], solvent [for Bleomycin]) as the treated tardigrades. For ionizing irradiations, tardigrades were exposed to a $^{137}$Cs γ-ray source (GSR-D1 irradiator, RPS Services Limited) at a dose rate of 12.74 Gy/min or 16 Gy/min for a total dose of 100 Gy or 1000 Gy. Samples were collected at different time points after the end of irradiation (for differential transcriptome analyses, at 4 hr post-irradiation; for differential proteomics, at 24 hr post-irradiation; for DNA damage and western blot analyses, at different time points from 0 hr to 7 days as indicated in the text). For investigation of early first time point just after IR, an electronic beam (KINETRON, GCR-MeV) with 4.5 MeV energy and maximum dose rate of 4.103 Gy/s was also used (*Lansonneur et al., 2019*). For Bleomycin treatment, after separating tardigrades from *Chlorella* by filtration, Bleomycin sulfate (#B5507, SIGMA) was added to the water at a concentration of 100 μM for 4 or 5 days. The response to Bleomycin treatment was examined in *H. exemplaris* and *A. antarcticus* but not in *P. fairbanksi* tardigrades (which grow more slowly than *H. exemplaris* and *A. antarcticus*).

## Production of antibodies against *H. exemplaris* proteins

Antibodies were raised against *H. exemplaris* proteins in rabbits by injecting selected peptide sequences by Covalab (Bron, France). For *He*-Ku80 (XRCC5), *He*-Ku70 (XRCC6 C-term), *He*-Dsup, *He*-TDR1, two peptides were injected in two rabbits and serums tested by western blot on *H. exemplaris* extracts from animals treated with 100 μM Bleomycin for 4 days. Serum showing the best response on day 88 after injections was purified on Sepharose beads coupled to immunogenic peptides. Peptides used were the following: *He*-TDR1: Peptide 1 (aa 37–52, C-IQDEVLDSSRSGSRNVcoNH2), Peptide 2 (aa 109–123, C-DKKKQKSLPKIRRDN-coNH2); *He*-Ku80 (XRCC5): Peptide 1 (aa 120–135, C-IQFDEESSKKKRFAKR-coNH2), Peptide 2 (aa 444–457, C-LDGKAKDTYQPNDE-coNH2); *He*-Ku70 (XRCC6 Cterm): Peptide 1 (aa 182–197, C-IRPAQFLYPNEGDIRG-coNH2), Peptide 2 (aa 365–37, C-YDPEGAHTKKRVYEK-coNH2); *He*-Dsup: Peptide 1 (aa 63–77, C-KTAEVKEKSKSPAKE-coNH2), Peptide 2 (aa 166–181, C-KEDASATGTNGDDKKE-coNH2). Production of antibody to *He*-phospho-H2AX is detailed in *Figure 1—figure supplement 1*.

## Western blot analysis

For each experiment, more than 10,000 *H. exemplaris* tardigrades were irradiated or untreated, and 1000–2000 tardigrades were collected at different time points after irradiation (30 min, 4 hr, 8h30, 24 hr, and 73 hr), centrifuged at 8000 rpm in 1.5 mL tubes for 5 min and the pellet frozen at –80°C until analysis. Lysis was carried out by sonication for 15 min (15 s ON/15 s OFF, medium intensity, Bioruptor, Diagenode) at 4°C in 100 μL/5000 tardigrades pellet of the following solution: 12 mM sodium deoxycholate, 12 mM N-Lauryl sarcosine sodium salt, 80 mM Tris-HCl pH 8.5, 400 mM NaCl, 1× cOmplete Protease Inhibitor (4693116001, Roche) and 1× PhosSTOP (PHOSS-RO, Roche). 0.4 vol of protein gel loading buffer (LDS 4×, Bio-Rad) and 0.1 vol of DTT 1 M was added and the mixture heated at 95°C for 5 min before loading onto Any kD Mini-PROTEAN TGX Stain-Free protein gel (4568124, Bio-Rad) and migration in 1× Tris-Glycine-SDS buffer at 200 V. Semi-dry transfer of proteins was performed with Transblot Turbo (Bio-Rad) onto nitrocellulose membrane and the membrane was cut half to separately detect proteins >50 kDa and <50 kDa. Protein detection was done with rabbit primary antibodies diluted 1:1000 or 1:2000 (20–200 ng/mL depending on antibody) in TBS-0.1% Tween 20, 5% BSA, supplemented with 1:10,000 dilution of anti-mouse alpha-tubulin (Clone B-5-1-2; SIGMA) for 1–3 hr at room temperature. Membranes were washed three times for 5 min with 1× TBS 0.1% Tween 20. Secondary antibodies diluted 1:5000 (anti-rabbit Starbright 700 [12004161, Bio-Rad]

for specific tardigrade proteins and anti-mouse Starbright 520 [12005867, Bio-Rad] for alpha-tubulin detection) in TBS-1% milk were incubated on membrane for 1 hr at room temperature. Membranes were washed three times for 5 min with 1× TBS 0.1% Tween 20 and unsaturated fluorescent signal was acquired using Chemidoc MP imager (Bio-Rad) with identical settings for samples to be compared within an experiment.

## gDNA extraction and analysis

For each experiment, more than 60,000 *H. exemplaris* tardigrades were irradiated or untreated, and 8000–12,000 tardigrades were collected at different times after irradiation (30 min, 4 hr, 8h30, 24 hr, and 73 hr), centrifuged at 8000 rpm in 1.5 mL tubes for 5 min and the pellet frozen at –80°C until analysis. gDNAs were extracted using the Monarch HMW DNA extraction kit for Tissue from New England Biolabs (NEB) with the following modifications: Lysis buffer was supplemented with proteinase K before proceeding to lysis. Pellets were resuspended in 35 µL of lysis buffer and grinded on ice for 1 min. This step was repeating twice leading to a final volume of ≈ 125 µL. After grinding, lysis proceeded in three steps: (i) incubation of 15 min at 56°C under gentle agitation (300 rpm), (ii) incubation of 30 min at 56°C, and (iii) incubation of 10 min at 56°C after addition of RNAse A. Proteinase K and RNAse A were added at the concentration recommended by NEB. Proteins were next separated from the gDNA by adding 40 µL of protein separation solution. Samples were next centrifuged (20 min, 16,000×g, 20°C). gDNA was precipitated with two beads and next eluted from the beads with 100 µL of elution buffer. Extracted gDNAs were analyzed by electrophoresis on native (0.9% agarose/1× TAE) or denaturing (0.9% agarose/30 mM NaOH/1 mM EDTA) gels. Electrophoresis conditions were: 2h30min/60 V/20°C for native gels, and 15 hr/18 V/20°C for denaturing gel. Native gels were stained with ethidium bromide and denaturing gels with SyBR Green I.

## Immunohistochemistry of tardigrades

Immunohistochemistry protocol was derived from *Gross and Mayer, 2019*; *Gross et al., 2018*. 10,000 tardigrades irradiated or untreated were sampled (by batches of 1000 tardigrades) at different time points after irradiation (5 min, 4 hr, 24 hr, or 72 hr), heated in Volvic mineral water 5 min at 70°C to extend the tardigrade body and directly fixed with 4% formalin (15686, EMS) in 1× PBS-1% Triton X-100 (PBS-Tx) by adding 5× solution. Fixation was carried out for 1–3 hr at room temperature. After 1 hr of fixation tardigrade was pelleted by centrifugation 5 min at 8000 rpm and kept in 200 µL of fixative solution. The cuticle was punctured by sonication using Bioruptor (Diagenode) in 6×1.5 mL tubes at a time (5 pulses of 5 s ON/ 5 s OFF in medium position). After fixation samples were pelleted by centrifugation and washed with 1 mL of 1× PBS-1% Triton X-100 three times (>3 hr wash) and tardigrades were transferred to a round 96-well plate for transferring tardigrades under the stereo-microscope. Blocking was done in 200 µL of 5% BSA in 1× PBS-1% Triton X-100 for at least 1 hr and in-house primary antibody against phosho-H2AX (rabbit) (dilution 1:10) in blocking buffer was applied on tardigrades overnight or for 3 days at 4°C. Washes with PBS-Tx were done four times for several hours the next day transferring tardigrades to a new well filled with 200 µL of PBS-Tx. Secondary antibody anti-rabbit Alexa Fluor 488 F(ab')2 (A11070, Invitrogen, dilution 1:500) in PBS-Tx supplemented with 1% BSA was incubated overnight at room temperature. Washes with PBS-Tx were done four times for several hours the next day transferring tardigrades to a new well filled with 200 µL. The last wash was done without Triton X-100. Hoechst 33342 4 µM in PBS 1× was incubated for 30 min and tardigrades were quickly transferred to water and to slide with minimum amount of liquid and finally mounted with ProLong Glass Antifade Mountant (P36982, Invitrogen).

For analysis of EdU staining, EdU in 20 mM HEPES-NaOH pH 7.5 was added to Volvic mineral water with 1000 filtered tardigrades at 50 µM 2 hr before irradiation or with control, untreated tardigrades and kept until 7 days post-irradiation. Samples were then processed as in *Gross et al., 2018*, except for the permeabilization of tardigrades which was carried out by sonication as for phospho-H2AX labeling.

Imaging was done by confocal microscopy (Zeiss [LSM 880 and AiryScan module] with ×63 lens) using Zenblack software or Leica DMIRE2 inverted microscope with ×10 lens and Metamorph software. Confocal Z-stacks and Maxprojections were processed and adjusted in Fiji ImageJ (v2.9.0). Images were treated with ImageJ software. Image panels were assembled and labeled in Microsoft PowerPoint for Mac (v16.66.1).

## RNA sequencing

15000–20,000 *H. exemplaris* (n=3), 1000–1500 *A. antarcticus* (n=3), and 500–1000 *P. fairbanksi* (n=4) for each independent biological sample were collected and subjected to IR treatments: (i) control animals non-irradiated and extracted 4 hr post-irradiation and (ii) irradiated animals (with $^{137}$Cs γ-ray source [GSR-D1 irradiator, RPS Services Limited] at a dose rate of 16 Gy/min) extracted 4 hr post-IR. 15,000–20,000 *H. exemplaris* and 1000–1500 *A. antarcticus* (three independent biological samples for each) were also subjected to Bleomycin treatment: (iii) control animals kept for 5 days in Volvic water and (iv) treated with 100 µM Bleomycin in Volvic mineral water for 5 days. After treatments, tardigrades were collected and washed by filtration on 40 µm nylon mesh and transferred to 1.5 mL tubes to pellet by centrifugation at 5 min at 8000 rpm. RNA was extracted using TRIzol (15596-026, Invitrogen) and by three freeze-thaw cycles in an ethanol-dry ice bath and mechanical disruption with glass beads and a plastic micro-tube homogenizer at each cycle. Yield was approximately 1 µg RNA/1000 tardigrades. Integrity of RNAs was checked on an Agilent 2100 Bioanalyzer with the Eukaryote Total RNA Nano kit and only samples with RNA Integrity Number >6 were sequenced. For *H. exemplaris* RNA samples, single-end (1×75) sequencing (TruSeq Stranded) was done on Illumina NextSeq 500 System. For *A. antarcticus* and *P. fairbanksi* (whose genomes are not available), paired-end (2×150) sequencing (TruSeq Stranded) was performed. In addition to short read RNA sequencing in the different experimental conditions, long read sequencing of a mixture of RNA samples of *A. antarcticus* and of RNA samples of *P. fairbanksi* species were performed with ONT to help improve transcriptome assembly. 1D libraries were prepared according to ONT protocol with 1D PCR Barcoding kit and full-length non-directional sequencing was performed on PromethION instrument (using Clontech-SMART-Seq v4 Ultra Low Input kit). Basecalling was conducted using guppy version (v6.4.2; parameters: `--min_qscore` 7 `--flowcell` FLO-MIN106 `--kit` SQK-PBK004 `--use_quantile_scaling` `--trim_adapters --detect_primer --trim_primers`).

## De novo transcriptome assembly

De novo transcriptome assembly was performed using full-length cDNA sequences for *A. antarcticus* and *P. fairbanksi*. We used RNA-Bloom (v2.0.0; *Nip et al., 2020*), to assemble the long reads, also using a subset of the produced short reads to correct the contigs. Then we used MMSeqs2 easy-cluster (v; parameters: `--min-seq-id` 0.85c 0.25 `--cov-mode` 1) to cluster together transcript isoforms to a gene (*Mirdita et al., 2019*). We set the minimum sequence identity to 0.85 and minimum coverage to 0.25 for both transcriptomes. Because *P. fairbanksi* is triploid with a high level of heterozygosity, we manually clustered differentially expressed genes that were annotated for the same function by EggNOG (see below). We aligned the isoforms from two or more clusters with the same EggNOG annotation using mafft (v1.5.0; *Katoh and Standley, 2013*) and we visually inspected the alignments on Geneious Prime (v2023.1). When isoforms from two or more clusters were properly aligning together, they were merged. For *A. antarcticus* and *P. fairbanksi*, we conducted the gene expression analysis using the softwares embedded in the Trinity suite (v2.15.0; *Haas et al., 2013*). We first mapped RNA sequencing reads on the transcriptomes using Salmon (statistics of the mapping of RNA sequencing reads are given in *Supplementary file 8*; *Patro et al., 2017*), then we measured differential gene expression using DESeq2 (*Love et al., 2014*). For *H. exemplaris,* as the genome was available, the gene expression analysis was conducted using Eoulsan workflow version 2.2 (v2.2-0-g4d7359e, build0 build on 764feac4fbd6, 2018-04-17 15:03:09 UTC) (*Lehmann et al., 2021*). We first mapped RNA sequencing reads on the de novo transcriptomes using STAR (*Dobin et al., 2013*), then we measured differential gene expression using DESeq2. The results were plotted using R (v4.2.2) with the ggplot2, ggrepel, and VennDiagram packages. Heatmap was plotted using GraphPad Prism (v9.3.1).

To annotate expressed genes from the three species, we ran EggNOG mapper (v2.1.9) on the assemblies using the 'genome' mode (*Cantalapiedra et al., 2021*). We also annotated all expressed genes through a sequence homology search against *Drosophila melanogaster* (GCF_000001215.4), *Caenorhabditis elegans* (GCF_000002985.6), *Homo sapiens* (GCF_000001405.40), *H. exemplaris* (GCA_002082055.1), *Paramacrobiotus metropolitanus* (GCF_019649055.1), and *R. varieornatus* (GCA_001949185.1). Since *H. exemplaris* genome is annotated, we ran the homology search against the target proteomes using blastp (v2.14.0). For *A. antarcticus* and *P. fairbanksi*, we conducted the homology search using the transcript as query (blastx) and as target (tblastn). Only blast hits with an e-value<0.05 were kept as potential homologs.

To identify tardigrade-specific genes, we ran a homology search using diamond (v2.1.6.160) (**Buchfink et al., 2021**) on the complete nr database (downloaded April 12 11:17:28 2023) for each transcript from *A. antarcticus* and *P. fairbanksi* (by blastx) or for each protein sequence for *H. exemplaris* (by blastp). Sequences with no-hit on the nr database (diamond blastx or blastp –e 0.001 `--taxon-exclude` 42241 `--ultra-sensitive`) and no hit in the previous annotation using proteomes of *C. elegans*, *D. melanogaster*, and *H. sapiens* (reciprocal hit –blastx and tblastn) were considered as tardigrade-specific and noted 'TardiSpe' in supplementary tables and data (**Mapalo et al., 2020**; **Hara et al., 2021**; **Kamilari et al., 2019**). Additionally, for TDR1 we conducted a blast search on the nr database using NCBI blastp, which is more sensitive but slower, than diamond blastp. In contrary to diamond, NCBI blastp produced multiple hits but on non-ecdysozoans organisms only (see **Supplementary file 3**). Similar results were obtained using HMMER (hmmsearch on EMBL-EBI website) on the reference proteome database (see **Supplementary file 3**).

## Proteome analysis

For each replicate (n=4 independent biological samples), 18,000 tardigrades for each of the four experimental conditions: (i) untreated; (ii) treated with Bleomycin at 100 µM for 4 days; (iii) irradiated (with $^{137}$Cs γ-ray source [GSR-D1 irradiator, RPS Services Limited] at a dose rate of 12.74 Gy/min) and collected after 4 hr; (iv) irradiated and collected after 24 hr. The tardigrades were split into two samples, with 13,000 tardigrades for differential proteomic analysis and 5000 tardigrades for western blotting experiments, that were pelleted by centrifugation in 1.5 mL tubes (8000 rpm for 5 min). The pellets were frozen at –80°C until all samples were available. All samples were lysed the same day 2 weeks before proteomics analysis in 100 µL iST-NHS-Lysis buffer (PreOmics GmbH) by sonication (Bioruptor Diagenode, 15 s ON/15 s OFF for 15 min), and heating at 95°C for 10 min. Soluble fractions were collected by centrifugation at 13,000×*g* for 15 min at 4°C and frozen at –80°C until analysis. Protein concentration in each sample was measured using BCA assay (Sigma-Aldrich). 30 µg of each sample were then prepared using the iST-NHS kit (PreOmics). Peptides resulting from LysC/trypsin digestion were labeled using TMTpro 16plex Label Reagent Set (Thermo Fisher Scientific) before mixing equivalent amounts for further processing. The peptide mix was then fractionated using the Pierce High pH Reversed-Phase Peptide Fractionation Kit (Thermo Fisher Scientific). The eight obtained fractions were analyzed by online nanoliquid chromatography coupled to MS/MS (Ultimate 3000 RSLCnano and Q-Exactive HF, Thermo Fisher Scientific) using a 180 min gradient. For this purpose, the peptides were sampled on a precolumn (300 µm × 5 mm PepMap C18, Thermo Scientific) and separated in a 200 cm µPAC column (PharmaFluidics). The MS and MS/MS data were acquired by Xcalibur (v2.9, Thermo Fisher Scientific). The mass spectrometry proteomics data have been deposited to the ProteomeXchange Consortium via the PRIDE (**Perez-Riverol, 2022**) partner repository with the dataset identifier PXD043897.

Peptides and proteins were identified and quantified using MaxQuant (v1.6.17.0, **Cox and Mann, 2008**) and the NCBI database (*H. dujardini* taxonomy, 2021-07-20 download, 20957 sequences), the UniProt database (*Chlorella* taxonomy, 2021-12-10 download, 21 219 sequences), and the frequently observed contaminant database embedded in MaxQuant (246 sequences). Trypsin was chosen as the enzyme and two missed cleavages were allowed. Peptide modifications allowed during the search were: C6H11NO (C, fixed), acetyl (Protein N-ter, variable), and oxidation (M, variable). The minimum peptide length and minimum number of unique peptides were respectively set to seven amino acids and one peptide. Maximum false discovery rates - calculated by employing a reverse database strategy - were set to 0.01 at peptide and protein levels. Statistical analysis of MS-based quantitative proteomic data was performed using the ProStaR software (**Wieczorek et al., 2017**). Proteins identified in the reverse and contaminant databases, proteins identified only in the *Chlorella* database, proteins only identified by site, and proteins quantified in less than three replicates of one condition were discarded. After log$_2$ transformation, extracted corrected reporter abundance values were normalized by variance stabilizing normalization method. Statistical testing for comparison of two conditions was conducted with limma, whereby differentially expressed proteins were sorted out using a Log$_2$ Fold Change cut-off of 0.3 and a limma p-value cut-off of 0.01, leading to an FDR inferior to 3% according to the Benjamini-Hochberg estimator.

## Production of recombinant *He*-TDR1 and *He*-TDR1-GFP

*He*-TDR1 and *He*-TDR1-GFP (see plasmid sequence in *Supplementary file 2*) were transformed in *E. coli* Rosetta 2(DE3). Single competent cells (Novagen, MerckMillipore). Protein expression was induced with 1 mM IPTG at $OD_{600}$=0.6–0.7 in 2xYT medium (containing 50 µg/mL carbenicillin, 35 µg/mL chloramphenicol, and 1% glucose) at 25°C during 20 hr. Cells were resuspended in lysis buffer 25 mM Tris-HCl pH 8, 500 mM NaCl, 20 mM imidazole, 1 mM TCEP (supplemented with protease inhibitor cocktail [Roche]), and lysed by sonication (Vibracell 75186- 7 s ON/7 s OFF, 50% amplitude, 10 min). The first step of purification was binding on Ni Sepharose 6 Fast Flow resin in batch (overnight at 4°C). After binding, the resin was washed with lysis buffer and the protein was eluted with 25 mM Tris-HCl pH 8, 500 mM NaCl, 250 mM imidazole, 10% glycerol, 1 mM TCEP. Eluted protein is concentrated (Amicon Ultra 10K) and diluted in buffer 25 mM Tris-HCl pH 8, 150 mM NaCl, 10% glycerol, 1 mM TCEP. The second step of purification was a gel filtration Superdex 200 increase 10/300 GL (Cytiva) equilibrated with 25 mM Tris-HCl pH 8, 150 mM NaCl, 10% glycerol, 1 mM TCEP using AKTA Pure instrument (Cytiva). Molecular weight calibration was obtained using Gel Filtration Standard (Bio-Rad).

## Protein-DNA interaction assays

For *He*-TDR1 interaction with plasmid DNA, a 5900 bp plasmid (a kind gift of *Xie et al., 2009*) circular or linearized at 20 ng/µL (i.e. 30 µM in bp was incubated with increasing amounts [0.625–10 µM with twofold serial dilutions]) of recombinant *HeTDR1* or in buffer containing 15 mM Tris-OAc pH 8, NaCl 180 mM, glycerol 2%, DTT 5 mM, BSA 0.1 mg/mL.

After 20 min binding at room temperature, samples were diluted twofold with sucrose 50% or sucrose 50% with proteinase K 80 U/µL and loaded onto a 0.75% agarose gel containing ethidium bromide. Migration was carried out for 35 min at 100 V (room temperature) and gel was imaged using GBox camera (Syngene).

For imaging of protein-DNA complexes, 1 µL of 5900 bp plasmid at 200 ng/µL was added to 10 µL of 10 µM of *HeTDR1*-GFP in 10 mM Tris-HCl pH 8, 150 mM NaCl, 10% glycerol, and 1 mM TCEP (protein storage buffer) to allow 30 µM in bp (i.e. 5 nM in plasmid molecule) final concentration. After 10 min incubation at room temperature the reaction was observed in a Kova counting chamber using Leica DMIRE2 40× lens. Images were acquired using Coolsnap HQ camera run by Metamorph software and treated with ImageJ software.

## Expression of *He*-TDR1-mNeongreen in *H. exemplaris* tardigrades

Act-He-TDR1-mNeongreen (NG) and Act-mCherry expression plasmids were constructed by Gibson assembly with plasmid backbone from *Loulier et al., 2014* (see sequence in *Supplementary file 2*). Actin promoter sequences were amplified from *H. exemplaris* gDNA, *HeTDR1* cDNA from RNA of *H. exemplaris* adult tardigrades, and mCherry from a mCherry containing plasmid. *He-Act-HeTDR1*-GFP and Act5C-mCherry plasmids (2 µg/µL in milliQ water each) were co-injected in 20 starved *H. exemplaris* adults maintained in an in-house-made PDMS injection chamber using Quartz micropipets. After 1 hr of microinjection, animals are let to recover in Volvic mineral water for 15 min to 1h15. In order to get the plasmid into cells, tardigrades are next electroporated using NEPA21 Super Electroporator (Nepa Gene). Electric shock was carried out in 0.7× Optimem (Gibco, Thermo Fisher Life Sciences) with settings from *Tanaka et al., 2023*. Hoechst 33342 20 µM for 2 days or 40 µM for 1 day was also added to mineral water (Volvic, France) for live staining of the nucleus. Animals were immobilized using carbonated water and imaged by confocal microscopy (Zeiss [LSM 880 and AiryScan module] with ×40 and ×63 lens) using Zenblack software.

## Expression of TDR1-GFP fusion proteins in human U2OS cells

Expression plasmids for fusion proteins of GFP and tardigrade proteins were constructed by Gibson assembly into pEGFP-N1 (Clontech) of the tardigrade cDNA (obtained by gene synthesis from Integrated DNA) or ordered from TwistBiosciences. Full nucleotide sequences of fusion proteins are provided as supplementary information (*Supplementary file 2*). Plasmids were transfected into human U2OS cells (purchased from ATCC ref ATCC-HTB-96 and regularly tested for mycoplasma contamination) by Amaxa electroporation with Nucleofector Kit V (Lonza) and plated in six-well plates containing glass slides.

## Immunolabeling of phospho-H2AX foci in response to Bleomycin treatment and image analysis

Two days after transfection, Bleomycin sulfate-treated (treatment was for 1 hr with 1 µg/mL Bleomycin sulfate) or control cells were rinsed three times with PBS and fixed with 3.7% formaldehyde in PBS for 15 min at room temperature, rinsed three times with PBS, permeabilized with PBS, 0.5% Triton for 15 min, blocked with PBS, 0.1% Tween, 5% fetal calf serum and incubated for 1h30 with specific anti-GFP (1 in 200 dilution of GFP Chicken polyclonal #ab13970, Abcam) and anti-phospho H2AX (1 in 800 dilution of BW301, Merck) antibodies. After three PBS, 0.1% Tween washes, cells were incubated with secondary anti-chicken (Alexa Fluor 488 Donkey Anti-Chicken. Reference: 703-546-155, Jackson ImmunoResearch) and anti-mouse (Cy3 Goat Anti-Mouse. Reference: 115-167-003, Jackson ImmunoResearch) antibodies. After three PBS, 0.1% Tween washes, cells were incubated with Hoechst solution (11534886, Invitrogen) diluted 1/5000 in PBS, 0.1% Tween and mounted with ProLong Glass Antifade Mountant (P36982, Invitrogen). Cells were next imaged by confocal microscopy (Zeiss LSM 880) using Zenblack software and ×40 lens in AiryScan mode acquisition of 7×7 contiguous XY fields and a Z-stack of 30 images at 0.1 µm intervals. Z-stacks were maximum projected and analyzed with Zen Blue software (v2.3) to automatically segment nuclei (using Hoechst staining), identify GFP-positive nuclei, and count phospho-H2AX foci within each nucleus. When phospho-H2AX staining occupied more than a 1/3 of the nucleus surface, the number of foci was arbitrarily fixed as >400. Statistical significance of the difference in numbers of phospho-H2AX foci was measured with the non-parametric, rank-based Kruskal-Wallis test using GraphPad Prism (v9.3.1).

## SEM of *P. fairbanksi* adults and eggs

Adults and eggs specimens were fixed with 2.5% glutaraldehyde in Volvic mineral water for 1 hr and washed three times with distilled water. The adults were put in microporous capsules and the eggs were filtered on Isopore membrane filters. The samples were dehydrated in ethanol series (50%, 70%, 90%, and 100%). Then critical point (Leica CPD300, PTME MNHN) was used to dry them. Adults and membranes with eggs were deposited on carbon adhesive on the scanning electron microscope (SEM) stubs, coated with platinum (Leica EM ACE600 coater PTME MNHN), and examined using a SEM (Hitachi SU3500, PTME MNHN).

# Acknowledgements

We thank Dr V Gross (University of Kassel) for advice on whole-mount immunolabeling experiments, Gabriel Ramasamy (RADEXP facility, Institut Curie) for help with irradiation experiments and Nawel Cherkaoui (joint service unit CNRS UAR 3750 at Institute Pierre Gilles de Gennes) for manufacturing the brass mold for the PDMS injection chamber. Funding for the project was from Sorbonne Université (Projet Emergence, projet TardiGRaDe), INSERM, CNRS, MNHN, and ANR TEFOR (ANRII-INBS-0014). MA was funded by a doctoral fellowship from Ministère de de l'Enseignement Supérieur et de la Recherche (France) and from Fondation pour la Recherche Médicale. The work at GenomiqueENS core facility was supported by the France Génomique national infrastructure, funded as part of the 'Investissements d'Avenir' program managed by the Agence Nationale de la Recherche (contract ANR-10-INBS-0009). The proteomic experiments were partially supported by Agence Nationale de la Recherche under projects ProFI (Proteomics French Infrastructure, ANR-10-INBS-0008) and GRAL, a program from the Chemistry Biology Health (CBH) Graduate School of University Grenoble Alpes (ANR-17-EURE-0003). Computing was supported by the Plateforme de Calcul Intensif et Algorithmique PCIA, Muséum national d'histoire naturelle, Centre national de la recherche scientifique, UAR 2700 2AD, Paris, France.

# Additional information

## Funding

| Funder | Grant reference number | Author |
|---|---|---|
| Sorbonne Université | SU-19-3-EMRG-06 | Carine Giovannangeli<br>Anne De Cian<br>Jean-Paul Concordet |
| Agence Nationale de la Recherche | ANR-II-INBS-0014 | Carine Giovannangeli<br>Jean-Paul Concordet |
| Agence Nationale de la Recherche | ANR-10-INBS-0009 | Laurent Jourdren<br>Corinne Blugeon |
| Agence Nationale de la Recherche | ANR-10-INBS-0008 | Annie Adrait<br>Yohann Couté |
| Agence Nationale de la Recherche | ANR-17-EURE-0003 | Annie Adrait<br>Yohann Couté |

The funders had no role in study design, data collection and interpretation, or the decision to submit the work for publication.

## Author contributions

Marwan Anoud, Conceptualization, Software, Formal analysis, Validation, Investigation, Visualization, Methodology, Writing – original draft; Emmanuelle Delagoutte, Conceptualization, Resources, Validation, Investigation, Visualization, Methodology, Writing – review and editing; Quentin Helleu, Conceptualization, Data curation, Software, Formal analysis, Validation, Investigation, Visualization, Methodology, Writing – original draft, Writing – review and editing; Alice Brion, Validation, Investigation, Methodology; Evelyne Duvernois-Berthet, Software, Investigation, Methodology, Writing – review and editing; Marie As, Investigation, Visualization, Methodology; Xavier Marques, Sophie Heinrich, Geraldine Toutirais, Marc Geze, Investigation, Methodology; Khadija Lamribet, Catherine Senamaud-Beaufort, Corinne Blugeon, Investigation; Laurent Jourdren, Software, Investigation; Annie Adrait, Formal analysis, Investigation, Methodology; Sahima Hamlaoui, Ilaria Giovannini, Lorena Rebecchi, Resources; Giacomo Gropplero, Resources, Methodology; Loic Ponger, Software, Supervision, Methodology, Writing – review and editing; Yohann Couté, Data curation, Formal analysis, Investigation, Methodology, Writing – review and editing; Roberto Guidetti, Resources, Investigation, Writing – review and editing; Carine Giovannangeli, Conceptualization, Supervision, Project administration, Writing – review and editing; Anne De Cian, Conceptualization, Resources, Formal analysis, Supervision, Validation, Investigation, Visualization, Methodology, Writing – original draft, Project administration, Writing – review and editing; Jean-Paul Concordet, Conceptualization, Supervision, Funding acquisition, Validation, Writing – original draft, Project administration, Writing – review and editing

## Author ORCIDs

Evelyne Duvernois-Berthet http://orcid.org/0000-0003-1442-6603
Catherine Senamaud-Beaufort http://orcid.org/0000-0001-7074-0416
Laurent Jourdren http://orcid.org/0000-0003-4253-1048
Corinne Blugeon http://orcid.org/0000-0001-5820-8785
Yohann Couté https://orcid.org/0000-0003-3896-6196
Roberto Guidetti http://orcid.org/0000-0001-6079-2538
Anne De Cian http://orcid.org/0000-0002-1737-2543
Jean-Paul Concordet https://orcid.org/0000-0001-8924-4316

Reviewer #1 (Public review): https://doi.org/10.7554/eLife.92621.3.sa1
Reviewer #3 (Public review): https://doi.org/10.7554/eLife.92621.3.sa2
Reviewer #4 (Public Review): https://doi.org/10.7554/eLife.92621.3.sa3
Author response https://doi.org/10.7554/eLife.92621.3.sa4

# Additional files

## Supplementary files

• Supplementary file 1. Manual annotation of TDR1 gene correcting the *H. exemplaris* reference genome annotation. (a) Alignment of *H. exemplaris* genome assembly GCA_002082055.1 with cDNA sequence of *He*-TDR1 obtained from Oxford Nanopore Technology (ONT) long read sequencing and cDNA cloning showed that a portion of TDR1 sequence is missing in the current assembly. (b) Alignment of PacBio reads used for genome assembly with *H. exemplaris* genome assembly GCA_002082055.1 and *He*-TDR1 cDNA. A zoom on the missing sequence (boxed in orange) shows the poor quality of PacBio reads used for genome assembly at this locus, likely explaining the absence of the missing *He*-TDR1 cDNA sequence in the current genome assembly. PacBio reads (SRX2495681, *Yoshida et al., 2017*) were downloaded from NCBI, mapped with minimap2 (*Li, 2018*) and alignment visualization was performed with Geneious Prime (v2023.1). Blue and red dots respectively indicate mismatches and indels in the alignment. cDNA sequence of *He*-TDR1 is provided in *Supplementary file 2* and encodes for a 146 amino acid long protein.

• Supplementary file 2. Sequences of plasmids and proteins of this study.

• Supplementary file 3. BLAST and HMMER hit tables for He-TDR1 homologs.

• Supplementary file 4. List of tardigrade-specific proteins differentially expressed in all three conditions analyzed by mass spectrometry-based quantitative proteomics (4hr after irradiation, 24hr after irradiation and after Bleomycin treatment). Tardigrade-specific proteins are ranked according to the Log$_2$ Fold Change (from highest to lowest) at 4hr post-irradiation.

• Supplementary file 5. Genes of major DNA repair pathways of DNA damages caused by ionizing radiation (IR) are upregulated in all three species studies. (a) g:Profiler analysis of differentially expressed genes in *A. antarcticus* after IR. (b) g:Profiler analysis of differentially expressed genes in *P. fairbanksi* after IR. (c) Schematic representation of DNA repair genes up- or downregulated in *H. exemplaris* after IR. Genes in green colored boxes are upregulated with adjusted p-value<0.05. Genes in red colored boxes are downregulated with adjusted p-value<0.05. Genes with no homolog identified in *H. exemplaris* genome are checked with a black cross. (d) Table of DNA repair genes up- or downregulated in *H. exemplaris*, *A. antarcticus,* or *P. fairbanksi* after IR, classified based on the KEGG database. Note that the alternative end joining pathway, also called the micro-homology-mediated end joining (MMEJ) pathway is not currently included in the KEGG database. In the KEGG database, the POLQ gene is included in the BER pathway only. Only genes showing differential gene expression with adjusted p-value<0.05 are shown.

• Supplementary file 6. Phylogenomics of tardigrade-specific genes involved in resistance to desiccation and DNA damages. Green and white boxes indicate presence and absence, respectively, of the indicated gene or gene family as found in *Arakawa, 2022*, and in this work for TDR1. Light green indicates presence of potential *Rv*-Dsup ortholog with hypothetical function in radio-resistance (*Arakawa, 2022*). The figure in *Supplementary file 6* is adapted from Figure 3 of *Arakawa, 2022*, and augmented with additional information from this work. A TDR1 homolog could not be identified by BLAST analysis of *R. varieornatus* genome and available transcriptomics data. Sequence similarity of a potential TDR1 protein in *R. varieornatus* may be too low and indicate alternative mechanisms of radio-resistance in *R. varieornatus*, e.g., based on stronger activity of the *Rv*-Dsup compared to *He*- and *Aa*-Dsup. Investigation in additional species may help to clarify the presence/absence of TDR1 in the *Ramazzottius* genus.

• Supplementary file 7. Identification of *P. fairbanksi* tardigrades isolated and reared from moss garden. (a) Scanning electron microscope (SEM) of adult specimen with magnification of mouth and claws. (b) SEM of egg with magnification of characteristic spikes decorating the egg surface. Bright-field morphological analysis performed in parallel by one of the co-authors (R Guidetti) confirmed *P. fairbanksi* identification. Species identification was further confirmed by 28S, 18S, COX1, ITS2 sequencing (see next page). For further information on *P. fairbanksi*, see *Kayastha et al., 2023*.

• Supplementary file 8. Mapping of RNA sequencing reads statistics.

• MDAR checklist

## Data availability

As stated in the methods section, all sequencing data have been deposited in NCBI SRA under accession code Bioproject ID PRJNA997229 and all proteomics data have been deposited in the ProteomeXchange Consortium via the PRIDE (*Perez-Riverol, 2022*) partner repository with the dataset identifier PXD043897. TDR1 mRNA sequences of A. acutuncus, H. exemplaris and P. fairbanksi are

available from Genbank with accession numbers PP830927, PP830928 and PP830929, respectively. All data analysed during this study are included in the manuscript and supporting files; source data files have been provided for all figures. All materials generated in the paper are available from the authors upon reasonable request.

The following datasets were generated:

| Author(s) | Year | Dataset title | Dataset URL | Database and Identifier |
|---|---|---|---|---|
| Anoud M, Delagoutte E, Helleu Q, Brion A, Duvernois-Berthet E, As M, Marques X, Lamribet K, Senamaud-Beaufort C, Jourdren L, Adrait A, Heinrich S, Toutirais G, Hamlaoui S, Gropplero G, Giovannini I, Ponger L, Gèze M, Blugeon C, Coute Y, Guidetti R, Rebecchi L, Giovannangeli C, De Cian A, Concordet JP | 2023 | RNA-Seq analysis of Hybsibius exemplaris, Acutuncus antarcticus and Paramacrobiotus fairbanksi under DNA damaging stresses | https://www.ncbi.nlm.nih.gov/bioproject/?term=PRJNA997229 | NCBI BioProject, PRJNA997229 |
| Anoud M, Delagoutte E, Helleu Q, Brion A, Duvernois-Berthet E, As M, Marques X, Lamribet K, Senamaud-Beaufort C, Jourdren L, Adrait A, Heinrich S, Toutirais G, Hamlaoui S, Gropplero G, Giovannini I, Ponger L, Gèze M, Blugeon C, Coute Y, Guidetti R, Rebecchi L, Giovannangeli C, De Cian A, Concordet JP | 2023 | Proteomic analysis of global response to ionizing radiation in Hypsibius exemplaris | http://www.ebi.ac.uk/pride/archive/projects/PXD043897 | PRIDE, PXD043897 |
| Anoud M, Delagoutte E, Helleu Q, Brion A, Duvernois-Berthet E, As M, Marques X, Lamribet K, Senamaud C, Jourdren L, Adrait A, Heinrich S, Toutirais G, Hamlaoui S, Gropplero G, Giovannini I, Ponger L, Geze M, Blugeon C, Coute Y, Guidetti R, Rebecchi L, Giovannangeli C, De Cian A, Concordet J-P | 2024 | Acutuncus antarcticus isolate MNHN TDR1 (tdr1) mRNA, complete cds | https://www.ncbi.nlm.nih.gov/nuccore/PP830927 | NCBI GenBank, PP830927 |

*Continued on next page*

*Continued*

| Author(s) | Year | Dataset title | Dataset URL | Database and Identifier |
|---|---|---|---|---|
| Anoud M, Delagoutte E, Helleu Q, Brion A, Duvernois-Berthet E, As M, Marques X, Senamaud C, Jourdren L, Adrait A, Heinrich S, Toutirais G, Hamlaoui S, Gropplero G, Giovannini I, Ponger L, Geze M, Blugeon C, Coute Y, Guidetti R, Rebecchi L, Giovannangeli C, De Cian A, Concordet J-P | 2024 | Hypsibius exemplaris isolate Z151 TDR1 (tdr1) mRNA, complete cds | https://www.ncbi.nlm.nih.gov/nuccore/PP830928 | NCBI GenBank, PP830928 |
| Anoud M, Delagoutte E, Helleu Q, Brion A, Duvernois-Berthet E, As M, Marques X, Lamribet K, Senamaud C, Jourdren L, Adrait A, Heinrich S, Toutirais G, Hamlaoui S, Gropplero G, Giovannini I, Ponger L, Geze M, Blugeon C, Coute Y, Guidetti R, Rebecchi L, Giovannangeli C, De Cian A, Concordet J-P | 2024 | Paramacrobiotus fairbanksi isolate MNHN TDR1 (tdr1) mRNA, complete cds | https://www.ncbi.nlm.nih.gov/nuccore/PP830929 | NCBI GenBank, PP830929 |

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
