## [Editor Report · eLife assessment]

This study offers **valuable** insight into the remarkable resistance of tardigrades to ionizing radiation by showing that radiation treatment induces a suite of DNA repair proteins and by identifying a strongly induced tardigrade-specific DNA-binding protein that can reduce the number of double-strand breaks in human U2OS cells. The evidence of upregulation of repair proteins is **convincing**, and the case for a role of the newly identified protein in repair can be strengthened as genetic tools for tardigrades become better developed. The results will interest the fields of DNA repair and radiobiology as well as tardigrade biologists.

---

## [Referee Report · Reviewer #1 (Public review)]

Summary:

The manuscript aims to provide insights into the mediators and mechanisms underlying tardigrade radiation tolerance. The authors start by assessing the effect of ionizing radiation (IR) on the tardigrade lab species, H. exemplaris, as well as the ability of this organism to recover from this stress - specifically they look at DNA double and single strand breaks. They go on to characterize the response of H. exemplaris and two other tardigrade species to IR at the transcriptomic level. Excitingly, the authors identify a novel gene/protein called TDR1 (tardigrade DNA damage response protein 1). They carefully assess the induction of expression/enrichment of this gene/protein using a combination of transcriptomics and biochemistry - even going so far as to use a translational inhibitor to confirm the de novo production of this protein. TDR1 binds DNA in vitro and co-localizes with DNA in tardigrades.

Reverse genetics in tardigrades is difficult, thus the authors use a heterologous system (human cells) to express TDR1 in. They find that when transiently expressed TDR1 helps improve human cell resistance to IR.

This work is a masterclass in integrative biology incorporating a holistic set of approaches spanning next-gen sequencing, organismal biology, biochemistry, and cell biology. I think the importance of the findings is suitable and honestly, I find very little to critique in their experimental approaches.

Overall, I find this to be one of the more compelling papers on tardigrade stress-tolerance I have read.

---

## [Referee Report · Reviewer #3 (Public review)]

Summary:

This paper describes transcriptomes from three tardigrade species with or without treatment with ionizing radiation (IR). The authors show that IR produces numerous single strand and double strand breaks as expected and that these are substantially repaired within 4-8 hours. Treatment with IR induces strong upregulation of transcripts from numerous DNA repair proteins, and from the newly described protein TDR1 with homologs in both Hypsibioidea and Macrobiotoidea supefamilies. The authors show that TDR1 transcription produces newly translated TDR1 protein, which can bind DNA and co-localizes with DNA in the nucleus. At higher concentrations TDR appears to form aggregates with DNA, which might be relevant to a possible function in DNA damage repair. When introduced into human U2OS cells treated with the radiomimetic drug bleomycin, TDR1 reduces the number of double-strand breaks as detected by gamma H2AX spots. This paper will be of interest to the DNA repair field and to radiobiologists.

Strengths:

The paper is well-written and provides solid evidence of the upregulation of DNA repair enzymes after irradiation of tardigrades, as well as upregulation of the TRD1 protein. The reduction of gamma-H2A.X spots in U2OS cells after expression of TRD1 supports a role in a DNA damage.

Weaknesses:

Genetic tools are still being developed in tardigrades, so there is no mutant phenotype to support a DNA repair function for TRD1, but this may be available soon.

---

## [Referee Report · Reviewer #4 (Public Review)]

In this study, Anoud et al. show convincing results of genes involved in the radio-resistance of tardigrades. With transcriptomics, they found many genes involved in DNA repair pathways to be overexpressed after ionizing radiation. In addition, they found RNF146 coding for a ubiquitin ligase, and genes of the AMNP family. Finally, they more deeply characterized one upregulated gene that they named TDR1 (Tardigrade DNA damage Response 1) which seems specific to tardigrades. With proteomics they verified these results. They show that TDR1 binds DNA in vitro and co-localize with DNA in tardigrades. Because of the difficulties of carrying reverse genetics in tardigrades, the authors showed in vitro that human cells expressing TDR1 led to a reduced number of phospho-H2AX foci (indicating DNA damages) when treated with Bleomycin. Based on these results, the authors suggested that TDR1 interacts with DNA and might regulate chromosomal organization and favors DNA repair.

Strengths:

The paper provides solid evidence of the upregulation of DNA repair enzymes after irradiation of tardigrades, as well as upregulation of the TRD1 protein.

The reduction of gamma-H2A.X spots in U2OS cells after expression of TRD1 supports a role in a DNA damage.

The shown interaction of TDR1 with DNA.

Weaknesses:

No reverse genetics to support a DNA repair function for TRD1, even if I recognize that these remain difficult to carry in tardigrades.

No pulse field electrophoresis gels to show DNA damages in tardigrades, which remain apparently challenging to perform in tardigrades.

After revision, the manuscript gained in structure, and in precision.

Overall, the manuscript provides valuable and convincing results contributing to our knowledge of tardigrade radio resistance. While reverse genetics remain difficult to carry in tardigrades, the authors used the alternative approach to investigate TDR1 function in vitro in human cells.

This study illustrates integrative biology as it combines a set of different methodologies including next-generation sequencing, transcriptomic and proteomic analyses, immunohistochemistry, immunolabelling, in vitro assays and SEM. According to me, the quality and importance of the results make it of interest to the fields of DNA repair, radiobiology, and radio resistance.

---

## [Author Response]

The following is the authors’ response to the original reviews.

**Public Reviews:**

**Reviewer #1 (Public Review):**
Summary:The manuscript "comparative transcriptomics reveal a novel tardigrade specific DNA binding protein induced in response to ionizing radiation" aims to provide insights into the mediators and mechanisms underlying tardigrade radiation tolerance. The authors start by assessing the effect of ionizing radiation (IR) on the tardigrade lab species, H. exemplaris, as well as the ability of this organism to recover from this stress - specifically, they look at DNA double and single-strand breaks. They go on to characterize the response of H. exemplaris and two other tardigrade species to IR at the transcriptomic level. Excitingly, the authors identify a novel gene/protein called TDR1 (tardigrade DNA damage response protein 1). They carefully assess the induction of expression/enrichment of this gene/protein using a combination of transcriptomics and biochemistry - even going so far as to use a translational inhibitor to confirm the de novo production of this protein. TDR1 binds DNA in vitro and co-localizes with DNA in tardigrades.Reverse genetics in tardigrades is difficult, thus the authors use a heterologous system (human cells) to express TDR1 in. They find that when transiently expressed TDR1 helps improve human cell resistance to IR.This work is a masterclass in integrative biology incorporating a holistic set of approaches spanning next-gen sequencing, organismal biology, biochemistry, and cell biology. I find very little to critique in their experimental approaches.Strengths:(1) Use of trans/interdisciplinary approaches ('omics, molecular biology, biochemistry, organismal biology)(2) Careful probing of TDR1 expression/enrichment(3) Identification of a completely novel protein seemingly involved in tardigrade radio-tolerance.(4) Use of multiple, diverse, tardigrade species of 'omics comparison.Weaknesses:(1) No reverse genetics in tardigrades - all insights into TDR1 function from heterologous cell culture system.(2) Weak discussion of Dsup's role in preventing DNA damage in light of DNA damage levels measured in this manuscript.(3) Missing sequence data which is essential for making a complete review of the work.Overall, I find this to be one of the more compelling papers on tardigrade stress-tolerance I have read. I believe there are points still that the authors should address, but I think the editor would do well to give the authors a chance to address these points as I find this manuscript highly insightful and novel.

We thank the reviewer for his comments.

We agree that it will be important to further investigate the role of Dsup in radio-tolerance. We briefly mentioned this point in the discussion (p14). Our findings show that tardigrades undergo DNA damage at levels roughly similar to radio-sensitive organisms and therefore support a major role for DNA repair in the maintenance of genome integrity after exposure to IR. Nevertheless, we believe that more precise quantification of DNA damage may still reveal a contribution of genome protection to radio-tolerance of tardigrades compared to radio-sensitive organisms. Dsup loss of function experiments in tardigrades would clearly be the best way to assess this possibility. In the absence of experiments directly addressing the function of Dsup, we prefer to refrain from drawing any firm conclusion on prevention of DNA damage by Dsup and thus to keep a more open position. In any case, as discussed in the text, we note that Dsup has only been reported in Hypsibioidea and other molecular players, such as TDR1, are likely involved in radio-tolerance in other tardigrade species.

The sequence data can be accessed at the NCBI SRA database with Bioproject ID PRJNA997229.

**Reviewer #3 (Public Review):**
Summary:This paper describes transcriptomes from three tardigrade species with or without treatment with ionizing radiation (IR). The authors show that IR produces numerous single-strand and double-strand breaks as expected and that these are substantially repaired within 4-8 hours. Treatment with IR induces strong upregulation of transcripts from numerous DNA repair proteins including Dsup specific to the Hypsobioidea superfamily. Transcripts from the newly described protein TDR1 with homologs in both Hypsibioidea and Macrobiotoidea supefamilies are also strongly upregulated. They show that TDR1 transcription produces newly translated TDR1 protein, which can bind DNA and co-localizes with DNA in the nucleus. At higher concentrations, TDR appears to form aggregates with DNA, which might be relevant to a possible function in DNA damage repair. When introduced into human U2OS cells treated with bleomycin, TDR1 reduces the number of double-strand breaks as detected by gamma H2A spots. This paper will be of interest to the DNA repair field and to radiobiologists.Strengths:The paper is well-written and provides solid evidence of the upregulation of DNA repair enzymes after irradiation of tardigrades, as well as upregulation of the TRD1 protein. The reduction of gamma-H2A.X spots in U2OS cells after expression of TRD1 supports a role in DNA damage.Weaknesses:Genetic tools are still being developed in tardigrades, so there is no mutant phenotype to support a DNA repair function for TRD1, but this may be available soon.

We thank the reviewer for his comments.

**Reviewer #4 (Public Review):**
The manuscript brings convincing results regarding genes involved in the radio-resistance of tardigrades. It is nicely written and the authors used different techniques to study these genes. There are sometimes problems with the structure of the manuscript but these could be easily solved. According to me, there are also some points which should be clarified in the result sections. The discussion section is clear but could be more detailed, although some results were actually discussed in the results section. I wish that the authors would go deeper in the comparison with other IR-resistant eucaryotes. Overall, this is a very nice study and of interest to researchers studying molecular mechanisms of ionizing radiation resistance.I have two small suggestions regarding the content of the study itself.(1) I think the study would benefit from the analyses of a gene tree (if feasible) in order to verify if TDR1 is indeed tardigrade-specific.(2) It would be appreciated to indicate the expression level of the different genes discussed in the study, using, for example, transcript per millions (TPMs).Recommendations for the authors: please note that you control which revisions to undertake from the public reviews and recommendations for the authors

We thank the reviewer for his comments.

(1) To identify TDR1 homologous sequences in non-tardigrade species, we conducted extensive homology searches using multiple homology-based approaches (Blastp and Diamond against the NCBI non-redundant protein sequences (nr) database and hmmsearch against the EBI reference proteomes), which failed to identify TDR1 homologs in non-tardigrade ecdysozoans, thus strongly supporting that TDR1 is indeed tardigrade-specific.

To be clearer in the manuscript, we now state the absence of hits for TDR1 in non-tardigrade ecdysozoans. Given the absence of homologs in non-tardigrade species, it is not possible to make a gene tree with non-tardigrade species.

(2) To further document expression levels (which were already available from the Tables in the initial submission), we added MAplots (representing log2foldchange and logNormalized read counts) in the supplementary materials (Supp Figure 3 and Supp Figure 8). These additional figures clearly document that the DNA repair genes discussed in the main text and TDR1 are highly expressed genes after IR and after Bleomycin treatment.

**Recommendations for the authors:**

**Reviewer #1 (Recommendations For The Authors):**

We thank the reviewer for his comments.

(1) It has always seemed strange to me that tardigrades accumulate just as much DNA damage as any other organism when irradiated and yet their Dsup protein is supposed to shield and protect their DNA from damage. Perhaps this is an appropriate time for this idea to be reconsidered given the Dsup was NOT induced by IR in this study and the authors found that their animals incurred just as much damage as other biological systems. While Dsup is clearly not the focus of this manuscript, it is the protein most associated with tardigrade radio-tolerance and I would argue this new paper would call into question previous conclusions made about Dsup.

We agree that it will be important to further investigate the role of Dsup in radio-tolerance. We briefly mentioned this point in the discussion (p14). Our findings show that tardigrades undergo DNA damage at levels roughly similar to radio-sensitive organisms and therefore support a major role for DNA repair in the maintenance of genome integrity after exposure to IR. Nevertheless, we believe that more precise quantification of DNA damage may still reveal a contribution of genome protection to radio-tolerance of tardigrades compared to radio-sensitive organisms. Dsup loss of function experiments in tardigrades would clearly be the best way to assess this possibility. In the absence of experiments directly addressing the function of Dsup, we prefer to refrain from drawing any firm conclusion on prevention of DNA damage by Dsup and thus to keep a more open position. In any case, as discussed in the text, we note that Dsup has only been reported in Hypsibioidea and other molecular players, such as TDR1, are likely involved in radio-tolerance in other tardigrade species.

(2) While reverse genetics are difficult in tardigrades, they are not impossible, and RNAi can be used to good effect in these animals. In fact several authors on this manuscript have used RNAi to examine the necessity of genes in tardigrade stress tolerance in the past. Was an attempt made to RNAi TDR1? If not, why? With the large amount of work that the authors put into showing the sufficiency of TDR1 for increasing radiotolerance in cell culture, one would think looking at necessity in tardigrades would be of great interest. If RNAi was performed, what were the results? Even a negative result here is informative since a protein can be sufficient but not necessary for a function - if this were the case it would mean tardigrades have some redundant mechanism(s) for surviving radiation exposure beyond TDR1.

We have attempted RNAi experiments targeting TDR1 or a mix of DNA repair genes (including XRCC5) and examined response to a bleomycin treatment of 2 weeks. Unfortunately, we could not distinguish any difference between uninjected animals and animals injected with TDR1 dsRNAs , or the mix of DNA repair genes dsRNAs. We concluded that, bleomycin treatment, that we used because it is much easier to perform than irradiation, was perhaps not the best way to assay a potential impact of RNAi on survival since it required long term treatment for several days during which the effect of RNAi may have waned. Another attempt was therefore made injecting with TDR1 or control GFP dsRNAs and exposing animals to a 2000Gy IR treatment. We noticed that the viability was lower after injection with GFP dsRNAs than with TDR1 dsRNAs (likely due to problems we had with the injection needle during injections). The next day, animals were irradiated and we observed after 24h that animals injected with GFP dsRNAs exhibited higher lethality rates than animals injected with TDR1 dsRNAs or uninjected animals. We found that this set of experiments were not conclusive. Our current experimental set up will make it difficult to distinguish lethality due to injections from lethality due to potentially decreased resistance to IR. In particular, many key controls are difficult to make (in particular, we could not confirm the efficiency of target gene knockdown, as it is very challenging given the low amount of biological material available and the poor expression of these genes without irradiation). From a practical point of view, performing these experiments is thus very challenging. We nevertheless agree that, in future work, further experimentation is needed to examine the impact of knock-down by RNAi of TDR1 or of other genes such as DNA repair genes or Dsup, in tardigrade DNA repair and survival after IR. Gene knock-out with CRISPR-Cas9 is a very promising alternative to RNAi given that studies in mutant lines will eliminate the confounding effect of lethality due to injections.

(3) Regarding the U2OS experiments. I have several questions/points of clarification:a. Were survival/proliferation levels tested or only H2AX foci? I think that showing decreased H2AX foci (fewer double-stranded breaks) correlates with higher survival rates would be important.

In the experiments reported in Figure 6, cells were transiently transfected with expression vectors and we did not examine the impact on survival rates. U2OS cells are resistant to high doses of Bleomycin and testing survival would require longer exposure at much higher concentrations (Buscemi et al, 2014, PMID: 25486478). In order to try and better address an impact on cell survival, we therefore generated populations of cells stably expressing the candidate tardigrade proteins fused to GFP. Despite trying different experiment conditions for treatment with Bleomycin, we could not detect a reproducibly significant benefit on cell survival for any of the tardigrade proteins tested, including RvDsup which was used as a positive control (since it was previously reported to improve cell survival in response to X-rays). One possibility is that the analysis should be performed in clones and not in populations of cells with heterogeneous expression levels of the tardigrade protein tested. For example, expression levels of the tardigrade protein needed to reduce the number of phospho-H2AX foci in response to DNA damage may interfere with cell division. We note that in the original Dsup paper, the benefit of RvDsup on cell survival was reported in specific transgenic clones. Experiments in different biological systems have also started to document toxic effects of RvDsup expression, illustrating the challenge, when performing experiments in heterologous systems, to achieve suitable expression levels of the tested protein. Trying to perform such a finer analysis, in our opinion, would go beyond the scope of our manuscript and will be best addressed in future studies. We are therefore careful in the text not to make any claim on the benefit of TDR1 expression on cell survival in response to Bleomycin in human cultured cells.

(b) From the methods I am a bit confused as to how the images were treated/foci quantified. With the automatic segmentation and foci identification, is this done through the entire Z-series or a single layer? If the latter then I am not sure the results are meaningful, since we do not know how many foci might be present in other layers of the nuclei analyzed. If the former, please clarify this in the method since it is a very important consideration.

We have acquired images throughout the entire Z-series and edited the text to make it more clear ; We now write: “ Z-stacks were maximum projected and analyzed with Zen Blue software (v2.3)...”. To limit the time needed for image analysis, we have generated an artificial image by projecting the entire Z-series into a single image and counted foci in that single maximum projection image. Although there are potential drawbacks, such as potentially only counting one focus when two foci are superposed along the Z axis, this approach overcomes the limitations of quantification from a single layer. We further ensured statistical robustness of the analysis by performing quantification from several independent fields of the labelled cells and several independent biological replicates (n>=3 as now specified in the legend of figure 6a).

(c) RvDsup reduced levels of HXA1 foci in these experiments, however, HeDsup was not found to be enriched in the transcriptomic analysis performed here. Was there a reason HeDsup was not used in the cell-based experiments? One could argue that RvDsup is from a different species of tardigrade, but it is a bit concerning that an ortholog of a protein found NOT to be induced by radiation exposure seems to perform as well (if not better) than some versions of TDR1.

RvDsup is the protein initially shown to increase survival of human HEK293 cells treated with X-rays and reduce the number of phospho-H2AX foci induced: it was therefore used as a positive control in our experiments. The sequence of HeDsup is only poorly similar to RvDsup (with 26% identity) and activity of HeDsup in cultured cells has not been reported before. We therefore believe that HeDsup is not well suited to provide a positive control for the experiments performed in our manuscript.

(d) From the methods, it seems that cells were treated with Bleomycin and then immediately fixed without any sort of recovery time. In this short timeframe, the presence of TDR1 appears to be enough to deal with a substantial amount of double-stranded breaks (as evidenced by the reduced number of HXA1 foci). Does this make sense? How quickly could one expect DNA repair machinery to make significant progress in resolving damaged DNA? This response seems much faster than what was observed in tardigrades. Perhaps the authors to comment on this.

Kinetic studies in human cells show extremely rapid repair of DNA double-strand breaks. Sensing of DNA double strand breaks by PARP proteins takes place within seconds after irradiation by IR (Pandey and Black, 2021, PMID: 33674152). NHEJ is then observed to take place by formation of 53BP1 foci within 15 minutes (Schultz et al, 2000, PMID: 11134068). The number of phospho-H2AX and 53BP1 foci peaks at 30 minutes and starts declining thereafter, showing that at a significant number of sites, DNA repair is proceeding very rapidly (by NHEJ). Although we are not aware of any studies of DNA repair kinetics in U2OS cells after addition of Bleomycin, DNA damage must be instantaneous and further take place during exposure to the drug in parallel to DNA repair, which would be expected to have similar kinetics than after irradiation with IR.

In our experiments, several mechanisms may be involved in reducing the number of phospho-H2AX foci induced by Bleomycin, such as DNA protection (for Dsup expression) or stimulation of DNA repair (for RNF146 expression). For TDR1, the molecular mechanism involved remains to be determined. Given our finding that TDR1 can form aggregates with DNA, an additional possibility is that clustering of phospho-H2AX foci is induced.

(4) I could not find the sequences of the TDR1 proteins studied here. I did find the cDNA sequence of HeTDR1 in the final supplementary file, but not the other TDR1 orthologs. In the place where it appeared the TDR1 sequences from other tardigrades should be there were very short segments of the HETDR1 sequence. All sequences of proteins used in this study should be easily accessible to the reader and reviewers as it is not possible to review this work without accessing the sequences.

Our apologies for the inappropriate documentation of TDR1 sequences in the original manuscript. As requested, we have now included the TDR1 sequences in the Supplementary Table 4.

(5) Likewise, the RNA sequence data is said to be deposited in NCBI under PRJNA997229, but I do not find this available on NCBI.

The RNA sequence data was deposited in NCBI under the indicated reference before submission of the manuscript. The data has now been released and is fully available on NCBI.

(6) A few typographical errors: e.g., Page 10 - sentence 4 has two periods ". ." or page 14 which has an open parenthesis that is not closed.

These typos have been corrected in the revised manuscript.

**Reviewer #3 (Recommendations For The Authors):**

We thank the reviewer for his comments.

In Figure 4C, what fraction of the 50 genes upregulated in all species and treatments are DNA repair genes? Is there any other notable commonality between these 50 genes? The bulk of upregulated genes are specific to a species and to treatment with IR or bleomycin. What fraction of DNA repair genes are specific to a species or treatment?

The results in Figure 4C on the 50 putative orthologous genes upregulated in all species and treatments are further detailed in supp Figure 10. The legend to supp Figure 10 now provides the requested information: 14/50 genes are DNA repair genes and the other notable commonality is that 21/50 are “stress response genes”. We did not further breakdown the analysis to evaluate the fraction of DNA repair genes specific to a species or treatment. It will be interesting to gather data in more species to hed light on the evolutionary history of DNA repair gene regulation in response to IR.

How does the suite of upregulated tardigrade DNA repair proteins after IR or bleomycin compare with DNA or repair proteins upregulated under similar treatments in human cells? Are they quantitatively or qualitatively different, or both?

There is a great wealth of studies documenting genes differentially expressed in human cells in response to IR (e.g. Borras-Fresneda et al, 2016, PMID: 27245205; Rieger and Chu, 2004, PMID: 15356296; Budwoeth et al, 2012, PMID: 23144912 ; Rashi-Elkeles et al, 2011, PMID: 21795128; Jen and Cheung, 2003, PMID: 12915489...). Upregulation of DNA repair and cell cycle genes is commonly found. However, the number of DNA repair genes induced is always very limited and fold stimulation very modest compared to the massive upregulation observed in tardigrades.

On page 14, please explain the acronym BER. Do the authors mean Base Excision Repair? Or something else?

As assumed by the reviewer, the acronym BER stands for Base Excision Repair. The acronym has been removed from the main text and replaced by the full name.

**Reviewer #4 (Recommendations For The Authors):**

We thank the reviewer for his comments.

Abstract:The abstract is fine. What was hard to grasp at the beginning is why TDR1 gene was named that way. It should be clearer that this study decided to further focus on that gene, one of the most overexpressed gene after IR, with an unknown function. Then maybe introduce that it was found to be unique to tardigrade and to interact with DNA. Therefore, it was named TDR1.Introduction:

The introduction has been modified according to the suggestions of Reviewer#4 below. One of the suggested references, Nicolas et al 2023 from the Van Doninck lab, was published while our manuscript was under review and cannot be considered as background information for our study.

1st paragraph:The study is on tardigrades, I found it strange that the first paragraph is on D. radiodurans. I think it is fine to mention what is known in bacteria and eucaryotes but we should already know what will be the main topic in the first paragraph of the introduction. Some details about D. radiodurans seem less important and distracting from the main topic (3D conformation).2nd paragraph:When mentioning radio-resistant eurcaryotes the authors do not mention the larvae of the anhydrobiotic insect Polypedilum vanderplanki. Stating that the mechanisms of resistance are poorly characterized should perhaps be nuanced. There are some recent studies on D. radiodurans (Ujaoney et al., 2017) the insect P. vanderplanki (Ryabova et al., 2017), tardigrades (Kamilari et al., 2019), and rotifers (Nicolas et al., 2023, Moris et al., 2023). Perhaps these papers are worth indicating that if mechanisms are not elucidated yet, recent studies suggest some actors involved in their resistance. Regarding the sentence stating that DNA repair rather than DNA protection plays a predominant role in the radio-resistance of bdelloid rotifers should also be nuanced. Indeed, many chaperones, antioxidants were mentioned to play a role in the radio-resistance of bdelloid rotifers (Moris et al., 2023). The authors mentioned the reference Hespeels et al., 2023 which is not found in their list of references, I am not sure which paper they refer to. The last sentence of the second paragraph does not mean much. I am not sure what the authors want to state with this. Perhaps they should specify if they mean that the function of many other genes overexpressed after IR remains unknown.Still, in the second paragraph, the authors focus on rotifers. They also do not mention what is known in the insect P. vanderplanki, which should be added. They still do not mention tardigrades. I think it is nice to first start with eucaryotes and then focus on tardigrades but as I mentioned before it would help to understand the aim of the paper if the first paragraph mentioned briefly the tardigrades and then could go into detail in the third paragraph.3rd paragraph:The sentence starting "with over 1400 species" best to remove from it "but they can differ in their resistance" and start the next sentence with that.4th paragraph:Very clear, we finally understand what is the focus of the manuscript.5th paragraph:Very clear. The authors should mention the names of the three studied species. Here, A. antarcticus is missing. The sentence "Further analyses in H. exemplaris... showed that TDR1 protein is present and upregulated". The authors should mention in which conditions the protein is upregulated. In that paragraph the authors mention phospho-H2AX: it might be good to introduce its functions before in the introduction (it is mentioned in the second sentence of the results: best to move it to the introduction).Results:There are a few sentences in this section which rather discuss the results than describe them. I think the manuscript might gain in quality if these interpretations of the results are moved into the discussion section. That would make the result section more concise and the discussion enriched.For instance, I suggest to move these sentences into the discussion:"the finding of persistent DSBs in gonads at 72h.... likely explains..."."suggesting that (i) DNA synthesis..."" Phospho-H2AX....also suggested""Moreover, expression of TDR1-GFP..., supporting the potential role of TDR1 proteins...""our results suggest that RNF146 upreguation could contribute...""AMNP gene g12777 was shown to increase...Based on our results, it is possible that..."Interpretations mentioned here above were always introduced cautiously (-"suggesting that (i) DNA synthesis..." ; -" Phospho-H2AX....also suggested" ; -"Moreover, expression of TDR1-GFP..., supporting the potential role of TDR1 proteins..." ; -"our results suggest that RNF146 upreguation could contribute..." ). These cautious interpretations were usually important in deciding next steps of the work. We therefore believe it is important to mention these interpretations in the results section to clearly expose the milestones marking the progression of the study.

For some results, they were directly discussed in the results section for the sake of concision (for example -"the finding of persistent DSBs in gonads at 72h.... likely explains..."; -"AMNP gene g12777 was shown to increase...Based on our results, it is possible that..." ) since, in our opinion, there was no need to mention them again in the main discussion.

Some other parts could be good to be moved into the introduction:"Previous studies have indicated that irradiation with IR increases expression of Rad51,..." none of the actors involved in DNA repair are mentioned in the introduction. Also, change resistant into resistance"A. antarcticus ..., known for its resistant to high doses of UV....

We have moved these parts to the introduction as recommended.

It was in O. areolatus.... that the first demonstration..."

This piece of information is somewhat anecdotical. We choose to keep it it here in the results section. This information on the radio-resistance of the species P. areolatus is only relevant at this specific step of the study because it encouraged us to consider that P. fairbanksi, which we isolated fortuitously, would be a good model species for studying radio-resistance of tardigrades.

Here are some additional comments/suggestions on the result section:1st sectionRemove the Gross et al., 2018 from the sentence "using confocal microscopy", it looks otherwise that these results are from their study, not yours.

We have changed the text to make it clear that this is indeed a finding of Gross et al which was previously made in non-irradiated tardigrades. We replicated this finding, which showed that the protocol was working appropriately, and that we could use this control result for comparison with irradiated animals. We apologize for this confusion.

The text now states: “Using confocal microscopy, we could detect DNA synthesis in replicating intestinal cells of control animals, as previously shown by (Gross et al. 2018).”

2nd sectionIt is confusing what has been found induced by IR and/or by Bleomycin.I think it might help if the authors first present what is induced after IR, then write if it is similar after Bleomycin. Especially since they start to do it in the first paragraph of that section. However, they only mention TDR1 in the second paragraph dedicated to Bleomycin treatment which is confusing as it is also overexpressed after IR. It is also not clear if RNF146 is also induced by Bleomycin.

As recommended, the text presents first what is induced after IR and then what is induced by Bleomycin in the following paragraph. When reporting results with Bleomycin, we have provided a global assessment of what is common to both treatments in Supp Figure 3 and in Supp Table 3. In this figure, we also specifically highlighted several key genes of DNA repair induced by both treatments. These are also mentioned in the text (p8) to illustrate the point that many key DNA repair genes are common to both treatments. We have now added RNF146 to that list as recommended.

Regarding TDR1, it is not clear when introduced in the text as "promising candidate" why it is the case. It is clear in the figures but perhaps the authors should explain why they chose these genes for further analyses: high log2foldchange and expression level for instance. Regarding that last comment, it would be interesting to have an idea about the expression level of the genes with high log2foldchange. In Figures 2, 3, and 4 the pvalue and log2foldchange are represented but not the expression level (ideally Transcript per Millions). These values would give an additional idea on the importance of that gene. While looking at the figures, it is unclear why you did not further characterize other genes with high log2foldchange (some with even hints of their function): the mentioned RNF146, macroH2A1 (not even mentioned in the results), some genes unannotated in the figures with likely unknown functions,

When selecting genes of interest, we did indeed take into account high expression levels. To more clearly document expression levels (which were already available from the Tables), we added MAplots (representing log2foldchange and logNormalized read counts) in the supplementary materials (Supp Figure 3 and Supp Figure 8).

It is also unclear at that stage why you named it "Tardigrade DNA damage response protein", as it is characterized as DNA repair/damage proteins by specific GO id or is it based on your downstream analyses, I think it might be worth to quickly mention the reason of that name.

The name illustrates two points which were already characteristic at this point in time of the study i.e. (1) it is a tardigrade specific protein and (2) it is induced in response to DNA damage.

Regarding the BLAST analyses the protein was searched in *C. elegans, D. melanogaster* and *H. sapiens*. Why only these three species? What were the threshold evalues used for these analyses. As mentioned in the main comment, it would be worth searching species phylogenetically close to tardigrades to verify if it is well-tardigrade specific. Did you try to make a gene tree, after looking for a conserved domain (using hmmersearch)?

As indicated in the methods section, the “Tardigrade-specific" annotation was determined by absence of hits after high-throughput alignment (with diamond using –ultrasensitive-option) on the NCBI nr database and absence of hits after blast search on *C. elegans*, *D. melanogaster* and *H. sapiens* proteomes as a complementary criterion (the latter blast search was primarily performed to enrich for functional annotations). Based on these criteria, TDR1 was annotated as “Tardigrade-specific”. As stated in the text, we also searched for TDR1 related sequences with (1) blastp (which is more sensitive than diamond) on the NCBI nr database and (2) HMMER on Reference Proteomes, and no hits were found among non-tardigrade ecdysozoans organisms, confirming TDR1 is specific to tardigrades. For Blast search for example, there were five hits in non-ecdysozoans organisms (two cephalochordates, one mollusc and two echinoderma). The blastp and HMMER results are now included in the revised supplementary material (Supp Table 5). These very few hits in species phylogenetically distant from tardigrades cannot be taken to support the existence of TDR1 genes outside tardigrades.

To be clearer in the manuscript, we now state the absence of hits for TDR1 in non-tardigrade ecdysozoans. Given the absence of homologs in non-tardigrade species, it is not possible to make a gene tree with non-tardigrade species.

Page 9: "Proteins extracts from H. exemplaris... at 4h and 24h..." I think this sentence can be removed as this is mentioned again 2 paragraphs after: "...we conducted an unbiased proteome analysis... at 4h..." The log2foldchange threshold mentioned for the proteomic analyses is 0.3: why this threshold, was it chosen randomly?

This is threshold is commonly used when considering log2foldchange with the technology used in our study, an isobaric multiplexed quantitative proteomic strategy which is known to compress ratios (Hogrebe et al. 2018).

Page 10:It would be good for more clarity to indicate at the beginning of the new section which species were investigated after IR or Bleomycin treatment.TDR1 homologs in the other tardigrade species were identified based on what? Best reciprocal hit?

As indicated in the methods section of the manuscript, we searched for homologs in other tardigrade species by BLAST. A best reciprocal hit approach was not performed to try to determine which homologs might be orthologs. In particular, most TDR1 homologs identified are known from transcriptome assemblies and high-contiguity genome assemblies are needed to more confidently identify orthology (using synteny). The results of the BLASTP search are now provided as supplementary material (Supp Table 5).

Preliminary experiments indicated that A. antarcticus and P. fairbanski survived exposure to 1000 Gy: is there a supplementary graph showing this?

We have corrected the text to avoid any confusion. We have not rigorously examined the dose-dependent survival of P. fairbanksi in response to irradiation. Text was changed to: “We found by visual inspection of animals after IR that A. antarcticus and P. fairbanksi readily survived exposure to 1000 Gy.”

Page 11:"A set of 50 genes was upregulated in the three species": please be precise if only after IR.

Done

These genes cannot be the same as they are from different species. Did the author mean that they are coding for similar proteins? It might be good to give some more details even if the supplementary figure is mentioned.

Obviously, these genes are putative orthologs. We have changed the text to:

” a set of 50 putative orthologous genes was upregulated in response to IR in all three species”

Discussion:General comment: the discussion is focused mainly on TDR1, it would be nice to also discuss the other results: DNA repair genes, RNF146.

A whole paragraph is devoted to discussion of results on DNA repair genes and RNF146. We have extended that discussion following on the suggestion of the reviewer. In particular, we have explicitly mentioned the apparent paradox that XRCC5 and XRCC6, which are among the most highly stimulated genes at the mRNA level, only display modest upregulation at the protein level. Although further studies would be needed to examine the mechanisms involved, we propose that upregulation of RNF146, whose human homolog has been shown to drive degradation of PARylated XRCC5 and XRCC6 proteins in response to IR (Kang et al. 2011), may be responsible for higher degradation rates and may thus counterbalance increased levels of protein synthesis.

Pulse field electrophoresis would be nice to be performed. It has been used to assess DSBs in bdelloid rotifers, is it possible in tardigrades?

As stated in the discussion, we believe that it would be challenging to perform pulse field electrophoresis in tardigrades. However, if possible, these experiments would certainly bring invaluable information to complement our analysis of DNA damage induced by IR.

"By comparative transcriptomics": please rephrase that sentence.Proteins acting early in DNA repair: I am not sure I understand this sentence. Actors as ligases act not at the beginning of the repair pathways.

Well noted. We have removed ligases from the list.

It is confusing that the authors mention NHEJ and double-strand break repair pathways as different pathways. There are 2 main pathways to repair DBSs: NHEJ and HR. It would be nice to add a reference to the sentence "PARP proteins act as sensors of DNA damage etc."

A typo in the sentence gave rise to the misleading suggestion that NHEJ is not a double strand repair pathway. It has been corrected.

A reference has been added for PARP proteins.

It would be nice if the authors can explain deeper their suggestion that degradation of DNA repair actors is essential for tardigrade IR resistance.

We have expanded this part of the discussion and hope that it is clearer.

“For XRCC5 and XRCC6, our studyestablished, by two independent methods, proteomics and Western blot analysies, that the stimulation at the protein level could be much more modest 6 and 20-fold at most (Supp Figure 6) than at the RNA level (420 and 90 fold respectively). This finding suggests that the abundance of DNA repair proteins does not simply increase massively to quantitatively match high numbers of DNA damages. Interestingly, in response to IR, the RNF146 ubiquitin ligase was also found to be strongly upregulated. RNF146 was previously shown to interact with PARylated XRCC5 and XRCC6 and to target them for degradation by the ubiquitin-proteasome system (Kang et al. 2011). To explain the lower fold stimulation of XRCC5 and XRCC6 at the protein levels, it is therefore tempting to speculate that, XRCC5 and XRCC6 protein levels (and perhaps that of other scaffolding complexes of DNA repair as well) are regulated by a dynamic balance of synthesis, promoted by gene overexpression, and degradation, made possible by RNF146 upregulation. Consistent with this hypothesis, we found that, similar to human RNF146 (Kang et al. 2011), He-RNF146 expression in human cells reduced the number of phospho-H2AX foci detected in response to Bleomycin (Figure 6).”

Page 15: Please add a reference for the sentence "Functional analysis of promotor sequences in transgenic tardigrades etc."

The reference has been added to fix this omission.

Material and Methods:Small comments:40 μm mesh: space missing100 μm mesh: space missing(for Bleomycin): parenthesis missingremove "as indicated in the text"The investigated time points after radiation need to be clearly stated in the method section. It is also unclear in the IR and Bleomycin section which tardigrades were treated with what. Not all were treated with Bleomycin.

The small comments above have been fixed in the revised version of the manuscript.

Page 21: please precise the coverage of the RNA sequencing

Statistics on mapping of RNAseq reads are now provided in Supp Table 10.

Page 22: Was any read trimming performed? Anything about the quality check of the reads?

Trimming was conducted using trimmomatic (v0.39) and quality check using FastQC (v. ?) This information has been added to the Methods section.

Were the analyses confirmed by a second approach: for instance, EdgeR? Deseq2 and EdgeR do not always have the same results. For more robust analyses it is advised to use both.

Differential transcriptome analyses were conducted with DESeq2 only. The robustness of our identification of differentially expressed genes in response to IR stems from performing comparative analyses in three different species, rather than from using two bioinformatics pipelines in a single species. We also note that benchmarking reported in the initial DEseq2 paper showed that identification of differentially expressed genes with large log fold changes (which, as reported in our manuscript, is characteristic of many DNA repair genes in response to IR) is very consistent between DEseq2 and EdgeR.

Figures:Figure 2: Legend vertical dotted line does not indicate log2foldchange value of 4 in all panels: it would be good to indicate for panels a and c as well.

Figure 2has been improved following on the suggestions of the reviewer. Dotted lines now show log2foldchange value of 2 in all panels (ie Fold Change of 4 as mentioned in the main text).

Figure 2C: There are a few points with high log2foldchange which are not annotated: was it because nothing was found in the blast research? If yes, it would be good to indicate their functions. If not, it would be good to mention in the discussion that there are some genes with still unknown functions which might play an important role in the resistance of tardigrades to IR.

The few points which are not annotated in figure 2c can now be found in Supp Table 3 Some of them have no hit in Blast search, some others such as BV898_09662 or BV898_07145 have hits on DNA repair genes as RBBP8/CtIP or XRCC6 respectively but are not annnotated as such by eggnog in KEGG pathway.

Figure 4C: Why not have included the response of P. fairbanski to bleomycin? I guess it was not done, but it is unclear in the results and methods sections.

P.fairbanksi response to bleomycin wasn’t assessed as we didn’t get enough animals to run the study. The method section has been modified to precise this point.